# Adversarial Task Up-sampling for Meta-learning

**Yichen Wu[1,2]\*  Long-Kai Huang[2]†  Ying Wei[1]†**
[1]City University of Hong Kong, [2]Tencent AI Lab
{wuyichen.am97, hlongkai}@gmail.com, yingwei@cityu.edu.hk

## Abstract

The success of meta-learning on existing benchmarks is predicated on the assumption that the distribution of meta-training tasks covers meta-testing tasks. Frequent violation of the assumption in applications with either insufficient tasks or a very narrow meta-training task distribution leads to memorization or learner overfitting. Recent solutions have pursued augmentation of meta-training tasks, while it is still an open question to generate both correct and sufficiently imaginary tasks. In this paper, we seek an approach that up-samples meta-training tasks from the task representation via a task up-sampling network. Besides, the resulting approach named Adversarial Task Up-sampling (ATU) suffices to generate tasks that can maximally contribute to the latest meta-learner by maximizing an adversarial loss. On few-shot sine regression and image classification datasets, we empirically validate the marked improvement of ATU over state-of-the-art task augmentation strategies in the meta-testing performance and also the quality of up-sampled tasks.

## 1 Introduction

The past few years have seen the burgeoning development of meta-learning, *a.k.a.* learning to learn, which draws upon the meta-knowledge learned from previous tasks (i.e., *meta-training tasks*) to expedite the learning of novel tasks (i.e., *meta-testing tasks*) with a few examples. A sufficient number and diversity of meta-training tasks are pivotal for the generalization capability of the meta-knowledge, so that (1) they cover the true task distribution (i.e., environment [4]) from which meta-testing tasks are sampled, discouraging learner overfitting [23] and (2) the meta-knowledge empowers fast adaptation via the support set for each task, avoiding memorization overfitting [44]. Notwithstanding up to millions of meta-training tasks in benchmark datasets [24, 31], real-world applications such as drug discovery [40] and medical image diagnosis [14] usually have access to only thousands or hundreds of tasks, which puts the meta-knowledge at high risk of learner and memorization overfitting.

While early attempts towards improving the generalization capability of the meta-knowledge revolve around regularization methods that limit the capacity of the meta-knowledge [11, 44], recent works on augmentation of meta-training tasks have shown a marked improvement [20, 40, 43]. The objective of task augmentation is to draw the empirical task distribution which is formed by assembling Dirac delta functions located in each meta-training task closer to the true task distribution. Consequently, achieving this objective requires a qualified task augmentation approach to simultaneously possess the following three properties: (1) *task-aware*: the augmented tasks comply with the true task distribution, being not erroneous to lead the meta-knowledge astray (tasks A and B in Figure 1a); (2) *task-imaginary*: the augmented tasks cover a substantial portion of the true distribution, embracing task diversity which task-awareness is nonetheless inadequate to guarantee (tasks C, D, and E in Figure 1b); (3) *model-adaptive*: the augmented tasks are timely in improving the current meta-knowledge, to which the meta-knowledge before augmentation struggles to generalize (task F in Figure 1c).

---

\*Part of the work was done when the author interned in Tencent AI Lab.
†Corresponding author: Long-Kai Huang and Ying Wei

36th Conference on Neural Information Processing Systems (NeurIPS 2022).

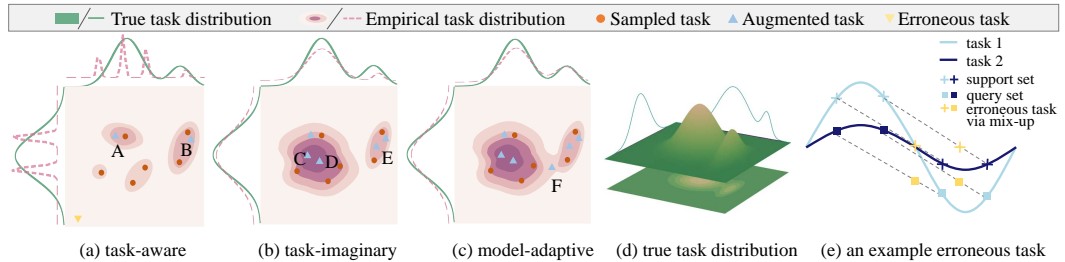

| | | | | | |
|---|---|---|---|---|---|
| (a) task-aware | (b) task-imaginary | (c) model-adaptive | (d) true task distribution | (e) an example erroneous task | |

Figure 1: Pictorial illustration of the three characteristics possessed by a qualified task augmentation approach, i.e., being (a) task-aware, (b) task-imaginary, and (c) model-adaptive. (d) shows the true task distribution that task augmentation aims to approximate, and (e) presents an example erroneous task violation of the true task distribution of sinusoidal functions $y = w\sin(x)$ ($w \in [0, 2]$).

Unfortunately, developing such a qualified task augmentation approach remains challenging. First, the task-aware methods sacrifice task diversity for task-awareness – they establish task-awareness by injection of the same random noise to labels of the support and query set [23], rotation of both support and query images [20], or mix-up of support and query examples within each task [40], all of which result in augmented tasks that are within the immediate vicinity of sampled meta-training tasks as shown in Figure 1a. Second, the task-imaginary method [43] that mixes up both support examples of two distinct tasks and their query examples in the feature space compromises on task-awareness – the resulting examples are even multi-modal in Figure 1e and constitute an erroneous task that fails to comply with the task distribution. Third, how to adaptively augment tasks that maximally improve the meta-knowledge and thereby the performance on meta-testing tasks remains unexplored.

To this end, we propose the Adversarial Task Up-sampling (ATU) framework to augment tasks that are aware of the task distribution, imaginary, and adaptive to the current meta-knowledge. Grounded on gradient-based meta-learning algorithms that are generally applicable to either regression or classification problems, ATU takes the initialization of the base learner as the meta-knowledge. Concretely, ATU consists of a task up-sampling network whose input is a task itself and outputs are augmented tasks. To ensure that the augmented tasks are imaginary and meanwhile faithful to the underlying task distribution, we train the up-sampling network to minimize the Earth Mover Distance between augmented tasks and the local task distribution characterized by a set of sampled tasks. Besides, we enforce the up-sampling network to produce challenging tasks that complement the current initialization, by maximizing the loss of the model adapted from the initialization on their query examples and minimizing the similarity between the gradient of the initialization with respect to their support examples and that of their query examples.

In summary, our main contributions are three-fold: (1) we present the first task-level augmentation network that learns to generate tasks that simultaneously meet the qualifications of being task-aware, task-imaginary, and model-adaptive; (2) we provide a theoretical analysis to justify that the proposed ATU framework indeed promotes task-awareness; (3) we conduct comprehensive experiments covering both regression and classification problems and a total of five datasets, where the proposed ATU improves the generalization ability of gradient-based meta-learning algorithms by up to 3.81%.

## 2 Related Work

As a paradigm that effectively adapts the meta-knowledge learned from past tasks to accelerate the learning of new ones, meta-learning has sparked considerable interest in many scenarios [38, 26, 36, 35], especially for few-shot learning. It falls into

Table 1: Summary of existing task augmentation strategies.

| Method | Task-aware | Task-imaginary | Model-adaptive |
|---|:---:|:---:|:---:|
| MetaAug [23] | ✓ | ✗ | ✗ |
| MetaMix [40] | ✓ | ✗ | ✗ |
| Meta-Maxup [20] | ✓ | ✗ | ✗ |
| MLTI [43] | ✗ | ✓ | ✗ |
| ATU | ✓ | ✓ | ✓ |

four major strands based on what the meta-knowledge is, i.e., optimizer-based methods [3, 37], feed-forward methods [24, 10, 39], metric-based methods [27, 29, 33] and gradient-based meth-

ods [9, 15, 42, 12, 6], where the inner optimizer, the mapping function from the support set to the task-specific model, the distance metric measuring the similarity between samples, and the parameter initialization are formulated as the meta-knowledge that enables quick adaptation to a task within a small number of steps. Our method is primarily evaluated on gradient-based methods which enjoy wide adoption and applicability in either classification or regression problems.

**Within-task Overfitting.** Few-shot learning puts meta-learning, especially gradient-based methods which require optimization of high-dimensional parameters within each task, at risk of within-task overfitting. Some works tackle the problem with various ways of reducing the number of parameters to adapt in the inner loop: only updating the head [22] or the feature extractor [21], learning data-dependent latent generative embeddings of parameters [25] or context parameters [48] to adapt, imposing gradient dropout [32], and generating stochastic input-dependent perturbations [13]. The other bunch of works alleviates the problem through data augmentation within each task. Ni et al. [20] applies standard augmentation like Random Crop and CutMix onto support samples. Sun et al. [28] and Zhang et al. [47] proposed to generate more data within a class via a ball generator and generative adversarial networks, respectively. These techniques designed for within-task overfitting, however, have been proved to lend little support to meta-overfitting which we focus on.

**Meta-overfitting.** Distinguished from traditional overfitting within a task, two types of meta-overfitting including memorization and learner overfitting have been pinpointed in [44, 23]. Despite meta-regularization techniques [11, 44] that limit the capacity of the meta-learner, task augmentation strategies [23, 19, 20, 16, 43, 40] have emerged as more effective solutions to meta-overfitting. Table 1 presents a summary of these strategies, except the strategies of large rotation [16] being part of Meta-Maxup [20] and DReCa [19] applicable to natural language inference tasks only. MetaAug [23] augments a task by adding a random noise on labels of both support and query sets, and MetaMix [40] mixes support and query examples within a task. Such within-task augmentation guarantees the validity of augmented tasks, i.e., being task-aware, though it almost does not alter the mapping from the support to the query set, i.e., generating limited imaginary tasks beyond meta-training tasks. Meta-Maxup [20] and MLTI [43] approach this problem via the cross-task mixup method, unfavorably at the expense of erroneous tasks. Our work seeks a novel task augmentation framework capable of generating tasks that not only meet the task-awareness and task-imagination needs but also adapt to maximally benefit the up-to-the-minute meta-learner.

## 3 Preliminaries

### 3.1 Meta-Learning Problem and Gradient-Based Meta-Learning

Meta-Learning model $f$ are trained and evaluated on episodes of few-shot learning tasks. Assume the task distribution is $p(\mathcal{T})$. A few-shot learning task $T_i$ i.i.d. sampled from $p(\mathcal{T})$ consists of a support set $D_i^s = (X_i^s, Y_i^s) = \{(x_{i,j}^s, y_{i,j}^s)\}_{j=1}^{K^s}$ and a query set $D_i^q = (X_i^q, Y_i^q) = \{(x_{i,j}^q, y_{i,j}^q)\}_{j=1}^{K^q}$, where $X_i^s$ and $Y_i^s$, ($X_i^q$ and $Y_i^q$) are the collection of inputs and labels in support (query) set, and $K^s$ ($K^q$) is the size of support (query) set.

The most representative gradient-based meta-learning algorithm is MAML [9]. MAML aims to learn an initialization parameter $\theta_0$ of the model $f$ that can be adapted to any new task after a few steps of gradient update. Concretely, given a specific task $D_i^s, D_i^q$ and a parametric model $f_\theta$, MAML initializes the model parameter $\theta$ by $\theta_0$ and updates $\theta$ by performing gradient descent on the support set $D_i^s$. It then optimizes the initialization parameter $\theta_0$ by minimizing the loss $\mathcal{L}$ estimated on the query set $D_i^q$. The objective of MAML can be formulated as

$$\min_{\theta_0} \mathbb{E}_{T_i \sim p(\mathcal{T})} \mathcal{L}(\phi_i, D_i^q), \qquad \text{s.t.} \quad \phi_i = \theta_0 - \alpha \nabla_{\theta_0} \mathcal{L}(\theta_0, D_i^s). \qquad (1)$$

### 3.2 Earth Mover's Distance

To estimate the distance between two tasks, we use Earth-Mover Distance (EMD). Earth-Mover Distance, a.k.a. Wasserstein metric, is a distance measure of two probability distributions or two sets of points, and is widely used in image retrieval and point cloud up-sampling works [46, 45]. Given two sets $S_1$ and $S_2$ with the same size, EMD calculates their distances as:

$$d_{EMD}(S_1, S_2) = \min_{\phi: S_1 \to S_2} \frac{1}{\|S_1\|} \sum_{s \in S_1} \|s - \phi(s)\|_2 \qquad (2)$$

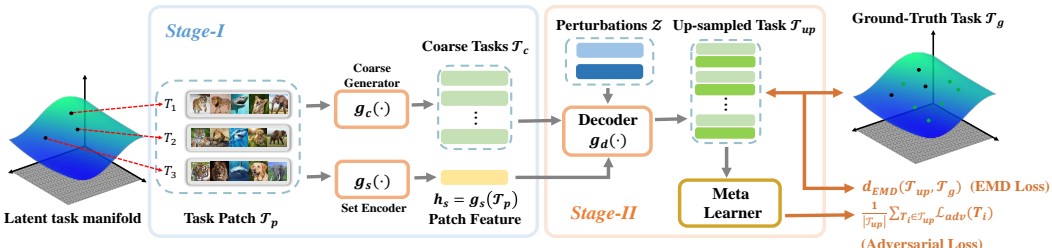

Figure 2: Illustration of the ATU algorithm with a task up-sampling network. The task up-sampling network consists of a set encoder $g_s(\cdot)$ which extracts a set feature of the input task patch, a coarse task generator $g_c(\cdot)$ which generates coarse tasks given the task patch, and a decoder $g_d(\cdot)$ which generates fine tasks from coarse tasks based on random perturbance and the set feature.

where $\phi$ is a bijective projection mapping $S_1$ to $S_2$. The value of EMD in (2) can be obtained by solving the linear programming problem w.r.t. $\phi$.

## 4    Adversarial Task Up-sampling

In practice, the task distribution $p(\mathcal{T})$ is unknown and we optimize the meta parameter $\theta_0$ with an empirical estimation of Eq. (1) over of meta-training tasks $\{T_i\}_{i=1}^{N_T}$ as

$$\min_{\theta_0} \frac{1}{N_T} \sum_{i=1}^{N_T} \mathcal{L}(\phi_i, D_i^q), \qquad \text{s.t.} \quad \phi_i = \theta_0 - \alpha \nabla_{\theta_0} \mathcal{L}(\theta_0, D_i^s) \qquad (3)$$

Given a finite set of meta-training tasks, the empirical task distribution may deviate from the true task distribution. The meta-model trained on such a finite set of tasks will cause memorization or learner overfitting [44, 23], which hurts the generalization to new tasks. To alleviate this problem, we propose a new task up-sampling network to generate a sufficient number of diverse tasks such that the empirical task distribution formed by the original meta-training tasks and the augmented tasks together is closer to the true task distribution. To achieve this, the tasks generated by the task up-sampling network should match the true task distribution and cover a large fraction of it. However, since the true task distribution and its underlying manifold are unknown, we cannot provide the task up-sampling network with explicit information about it. Instead, we generate new tasks by performing Task Up-sampling (TU) from a set of training tasks that implicitly comprise the latent task manifold information. The idea of Task Up-sampling is inspired by point cloud up-sampling methods [46, 45], which generate up-sampled points lying on the latent distribution (i.e., the shape) of the given local point patch. Similar to the point cloud up-sampling algorithm, our augmentation network receives a task patch consisting of a set of tasks $\mathcal{T}_p = \{T_i\}_{i=1}^{N_p}$ where $N_p$ is the set size, and generates up-sampled tasks $\mathcal{T}_{up} = \{\hat{D}^s, \hat{D}^q\}$ that are uniformly distributed over the same underlying task distribution as the task patch.

Due to the high complexity of task generation, it is infeasible to directly generate up-sampling tasks without sacrificing the quality of the tasks. Inspired by [46], we propose a two-stage generation strategy to generate the up-sampled tasks. In the first stage, we produce a sparse set of tasks, aiming at recovering the *global* task distribution of the task patch. The tasks obtained in the first stage are called coarse tasks. In the second stage, we generate multiple tasks for each coarse task, aiming to characterize the *local* task distribution around each coarse task. To guide the second generation stage, we use the patch features of the input task set as input to provide global task information and also multiple random noise vectors as input to provide directional perturbations to generate diverse tasks around the coarse task. The generation process is summarized in Fig. 2.

Our proposed task up-sampling network consists of 3 components, namely, a coarse task generator $g_c(\cdot)$, a set encoder $g_s(\cdot)$, and a decoder $g_d(\cdot)$. The coarse task generator is similar to a set auto-encoder. It first encodes the information of the whole input task set and then decodes it to generate $r_c N_p$ coarse tasks $\mathcal{T}_c = \{T_i^c\}$. The set encoder, denoted by $g_s(\cdot)$, extracts the set information of the input task patch as a patch feature $h_s$ to provide global information in the second-stage generation. For each task $T_i^c$ in $\mathcal{T}_c$, the decoder $g_d(\cdot)$ generates $r_d$ tasks located around $T_i^c$ in the task manifold

by taking as input $r_d$ random perturbations $\{z_i\}_{i=1}^r$ that are i.i.d. noise sampled from a uniform distribution and the set feature $h_s$ as input. In general, we use the same perturbations for each coarse task $T_i^c$ in $\mathcal{T}_c$. Finally, we obtain the up-sampled task set $\mathcal{T}_{up}$ consisting of $rN_p$ tasks as $\mathcal{T}_{up} = g_d(\mathcal{T}_c, \mathcal{Z}, h_s)$, where $r = r_c \times r_d$ is the up-sampling ratio. We denote the task augmentation network by $G_{\theta_g}(\mathcal{T}_p, \mathcal{Z})$ where $\theta_g$ is the trainable parameters of the task augmentation network.

In each iteration of the training phase, we construct $rN_p$ tasks to form the ground truth tasks set $\mathcal{T}_g$ (e.g., randomly select from the meta training task set). Then we sample $N_p$ tasks from $\mathcal{T}_g$ to form the task patch $\mathcal{T}_p$ and randomly sample $r$ perturbation noise vectors to form the perturbation set $\mathcal{Z} = \{z_i\}_{i=1}^r$. By feeding $\mathcal{T}_p$ and $\mathcal{Z}$ to the task augmentation network, we obtain the up-sampled task set $\mathcal{T}_{up} = G_{\theta_g}(\mathcal{T}_p, \mathcal{Z})$. To train the task augmentation network, we apply EMD loss between the up-sampled task set $\mathcal{T}_{up}$ and the ground-truth task set $\mathcal{T}_g$ to encourage the generated task set to have the same distribution as the true task distribution. However, it may still be insufficient to make the up-sampled tasks cover a significant fraction of the true task distribution. In this case, the up-sampling tasks provide limited additional information compared with the original meta-training tasks, and thus the meta-learner has restricted benefit from the generated tasks. To generate more informative tasks for the meta-learner, we want the generated tasks to be difficult for the current meta model $\theta_0$. Following [41], we measure the difficulty of the a task for $\theta_0$ by the loss estimated on query set w.r.t. to $\phi_i$, i.e. $\mathcal{L}(\phi_i, \hat{D}_i^q)$, and the gradient similarity between the support and query sets w.r.t. $\theta_0$, i.e. $\langle \nabla_{\theta_0}\mathcal{L}(\theta_0, \hat{D}_i^s), \nabla_{\theta_0}\mathcal{L}(\theta_0, \hat{D}_i^q)\rangle$. The large loss and small gradient similarity indicate a difficult task. Therefore, we want to maximize the following objective function to generate informative tasks:

$$\mathcal{L}_{adv}(\theta_0, (\hat{D}_i^s, \hat{D}_i^q)) = \eta_1 \mathcal{L}(\phi_i, \hat{D}_i^q) - \eta_2 \langle \nabla_{\theta_0}\mathcal{L}(\theta_0, \hat{D}_i^s), \nabla_{\theta_0}\mathcal{L}(\theta_0, \hat{D}_i^q)\rangle, \tag{4}$$

where $\eta_1$ and $\eta_2$ are two hyperparameters that control the strength of the two terms in $\mathcal{L}_{adv}$. We call this loss adversarial loss because it aims to increase the difficulty of the up-sampling tasks for the meta-learner while the meta-learner is trained to minimize the loss on the generated difficult tasks. And we named the proposed algorithm as Adversarial Task Up-sampling (ATU).

Together with the EMD loss, we obtain the objective to train the task up-sampling network:

$$\mathcal{L}_{ATU}(\theta_g, \mathcal{T}_p) = d_{EMD}(\mathcal{T}_{up}, \mathcal{T}_g) - \frac{1}{rN_p}\sum_{\hat{T}_i \in \mathcal{T}_{up}} \mathcal{L}_{adv}(\theta_0, (\hat{D}_i^s, \hat{D}_i^q)). \tag{5}$$

Note that the gradient of Eq. (5) will not be backpropagated to the meta model $\theta_0$ and the meta model will be updated by minimizing the meta loss in Eq. (3) on the up-sampled tasks $\mathcal{T}_{up}$. We summarize the proposed ATU in Algorithm 1 in Appendix A.

## 4.1 ATU on Regression and Classification Problem

Before introducing the details of regression and classification tasks, let us first review the Eq. (2), where $S_1$ and $S_2$ can be understood as the set of up-sampled tasks $\mathcal{T}_{up}$ and the set of ground-truth tasks $\mathcal{T}_g$, respectively. Each point $s$ in either set represents the embedding of a task.

**Regression Tasks.** We consider a simple regression problem: sinusoidal regression, which is widely used to evaluate the effectiveness of the meta-learning methods. In sinusoidal regression problem, before feeding a task $T_i = (D_i^s, D_i^q)$ to the task up-sampling network, we need to present the embedding of a sine regression task at first. In this paper, we combine all samples of the support set and query set as the embedding of a sine regression task, i.e., $s = [x_1^s, y_1^s, x_2^s, y_2^s, ..., x_{K^s}^s, y_{K^s}^s, x_1^q, y_1^q, x_2^q, y_2^q, ..., x_{K^q}^q, y_{K^q}^q] \in \mathbb{R}^{2(K^s + K^q)}$, where we sort the support set and query set such that $x_1^s \leq x_2^s \leq ... \leq x_{K^s}^s$ and $x_1^q \leq x_2^q \leq ... \leq x_{K^q}^q$. This sorting could make the task input is invariant to the permutation of data in support and query sets and thus the extracted feature of each task is permutation-invariant, which simplifies the design of the task up-sampling network. To generate the coarse tasks, we first use the set encoder $g_s(\cdot)$ to extract the set feature $h_s$ and directly generate the coarse tasks from the set feature $h_s$. Then we generate the up-sampled tasks $\mathcal{T}_{up}$ utilizing the decoder $g_d(\cdot)$ with perturbations $z$. Consequently, the dimension of generated tasks $\mathcal{T}_{up}$ is $(rN_p, 2(K^s, K^q))$, and that of ground truth tasks $\mathcal{T}_g$ is also $(rN_p, 2(K^s, K^q))$, where $r$ is the up-sampling ratio and $N_p$ is the set size of a task patch. For the regression problem, we add an extra EMD loss on the support and query set for each generated task $\hat{T}_i \in \mathcal{T}_{up}$ to encourage the points in the generated support set and query set to follow the same sinusoidal distribution, and the objective is shown in Eq. (6).

$$\mathcal{L}_{ATU}(\theta_g, \mathcal{T}_p) = d_{EMD}(\mathcal{T}_{up}, \mathcal{T}_g) + \eta_3 \frac{1}{rN_p} \sum_{\hat{T}_i \in \mathcal{T}_{up}} d_{EMD}(\hat{D}_i^s, \hat{D}_i^q) - \frac{1}{rN_p} \sum_{\hat{T}_i \in \mathcal{T}_{up}} \mathcal{L}_{adv}(\theta_0, (\hat{D}_i^s, \hat{D}_i^q)),$$

(6)

where $\hat{T}_i = (\hat{D}_i^s, \hat{D}_i^q)$ is obtained by transforming each of $\mathcal{T}_{up}$ back to a support set and a query set.

**Classification Tasks.** Dissimilar to regression tasks, classification tasks' labels of each class are randomly given under the mutually-exclusive setting [44]. For example, for an $N$-way $K^s$-shot classification problem with $K^q$ query samples for each class, the label $y$ in episodic-based meta-training is a randomly chosen value from $\{0, 1, ..., N-1\}$. In light of the fact that the label $y$ is not semantically meaningful, we only use the images $x$ to represent the embedding of an $N$-way classification task. Concretely, we reshape each task into a task pool of $(K^s+K^q)$ tasks, each of which is $N$-way 1-shot and represented as $s = [x_1, x_2, ..., x_N] \in \mathbb{R}^{Nd}$ where $d$ is the dimension of each image. Then, we can split the task into $(K^s+K^q)$ $N$-way 1-shot classification tasks without query examples. We represent the task by concatenating the input from $N$ classes in a fixed order (based on classes). We treat the $(K^s+K^q)$ tasks as a task patch and feed them to the task up-sampling network.

Since the task distribution of the image classification problem is extremely complex, it is impractical to generate the coarse tasks from a set feature. Instead, we use the original tasks as the coarse tasks and generate the up-sampled tasks by a more informative perturbance around the original tasks. To achieve this, we generate the perturbation by randomly sampling extra $K_M$ images from $N$ different classes in the base set. Then for the image $x_i$ in a class of a task in the coarse tasks $\mathcal{T}_c$, we subtract it from the $K_M$ images to obtain $K_M$ residual images and their corresponding set features (i.e., obtained from $g_s(\cdot)$), concatenate the set features with a noise vector, and use an attention network to obtain a residual images feature $x_i^{res}$ for the image $x_i$ given $K_M$ residual images. Finally, we generate the image as $x_i^u = x_i + x_i^{res}$ for the augmented task. We repeat this process for all images in a coarse task to get an augmented task and apply the $r$ noise vector to get $r$ up-sampled tasks. The dimension of generated tasks $\mathcal{T}_{up}$ and ground truth tasks $\mathcal{T}_g$ are both $(r(K^s + K^q), Nd)$, where $\mathcal{T}_g$ is obtained by mixing up with images in the memory bank. The whole training objective function is Eq. (5). More details of network structures and training details are shown in Appendix B.

## 5 Theoretical Analysis

We will introduce the formal definition of an up-sampled task that conforms to task-awareness, based on which we present the essential property of our proposed ATU framework in maximizing the task-awareness, compared to previous task augmentation approaches.

**Definition 1** (Task-aware Up-sampling). Suppose that we are given a set of $N_p$ tasks $\{\mathbf{X}_i, \mathbf{Y}_i\}_{i=1}^{N_p}$ from which we up-sample a new task $T_{up}$. For each $i$-th task, its ground-truth parameter that map the input $\mathbf{X}_i$ to the output $\mathbf{Y}_i$ is $\theta_i$, i.e., $\mathbf{Y}_i = f_{\theta_i}(\mathbf{X}_i)$. The up-sampled task $T_{up} = \{\mathbf{X}_u, \mathbf{Y}_u\}$ is defined to be task-aware, if and only if $\theta_u = g(\theta_1, \cdots, \theta_{N_u})$ and $\mathbf{Y}_u = f_{\theta_u}(\mathbf{X}_u)$ where $g$ is the up-sampling function and $\theta_u$ is the up-sampled parameter.

This definition states two prerequisites a task-aware up-sampling has to meet: (1) the up-sampling is performed in the functional space, which is to relate $N_u$ parameters via $g$; (2) the mapping between the input and the output of an up-sampled task satisfies $f_{\theta_u}$.

**Property 1** (Task-awareness Maximization). Consider $N_u = 2$, $g(\theta_1, \theta_2) = (1 - \lambda)\theta_1 + \lambda\theta_2$, $f_{\theta_1}(\cdot) = \mathbf{W}_1$, and $f_{\theta_2}(\cdot) = \mathbf{W}_2$. The proposed ATU algorithm that pursues an up-sampled task $T_{up} = \{\mathbf{X}_u, \mathbf{Y}_u\}$ via minimizing the EMD loss between $T_1$ and $T_2$ maximizes the task-awareness, i.e., minimizing the distance between $\mathbf{Y}_u$ and $f_{\theta_u}(\mathbf{X}_u)$.

*Proof.* According to the definition of EMD (Eq. (2)), it solves: $\phi^* = \arg\min_{\phi \in \mathbf{\Phi}} \sum_j \|\mathbf{x}_{1,j} - \mathbf{x}_{2,\phi(j)}\|_2$, where $\mathbf{\Phi} = \{\{1, \cdots, n_1\} \mapsto \{1, \cdots, n_2\}\}$ denotes the set containing all possible bijective assignments, each of which gives one-to-one correspondence between $T_1$ and $T_2$. Based on the optimal assignments $\phi^*$, the EMD is known to be defined as $d_{EMD} = \frac{1}{min\{n_1, n_2\}} \sum_j \|\mathbf{x}_{1,j} - \mathbf{x}_{2,\phi^*(j)}\|_2$. In light of the difficulty in mathematically formulating a possible up-sampled task $\tilde{T}_u$ that lies in the local manifold of $\{T_1, T_2\}$, we reasonably assume a simplified way of characterizing an up-sampled task $\tilde{T}_u$ to be $\tilde{\mathbf{y}}_{u,j} = \boldsymbol{\alpha}_{1,j}^T \mathbf{Y}_1 + \boldsymbol{\alpha}_{2,j}^T \mathbf{Y}_2$, $\tilde{\mathbf{x}}_{u,j} = \boldsymbol{\alpha}_{1,j}^T \mathbf{X}_1 + \boldsymbol{\alpha}_{2,j}^T \mathbf{X}_2$, $\forall j$, where each sample is a convex

combination of samples from both $T_1$ and $T_2$. The combination coefficients $\boldsymbol{\alpha}_{1,j}, \boldsymbol{\alpha}_{2,j} \in \mathbb{R}^{(K^s+K^q)\times 1}$, $\sum_k^{K^s+K^q} \alpha_{1,jk}=1, \sum_k \alpha_{2,jk}=1, \alpha_{1,jk}, \alpha_{2,jk} \geq 0, \forall k$. Different combination coefficients lead to a set of up-sampled task candidates $\{\tilde{T}_u\}$. We evaluate the task-awareness property of each candidate $\tilde{T}_u$, i.e., the distance between $\tilde{\mathbf{Y}}_u$ and $f_{\theta_u}(\tilde{\mathbf{X}}_u)$, to be $\|\tilde{\mathbf{Y}}_u - f_{\theta_u}(\tilde{\mathbf{X}}_u)\|_2 = \sum_j \|\tilde{\mathbf{y}}_{u,j} - f_{\theta_u}(\tilde{\mathbf{x}}_{u,j})\|_2 = \sum_j \|(\mathbf{W}_1 - \mathbf{W}_2)[\lambda \alpha_{1,j}^T \mathbf{X}_1 - (1-\lambda)\alpha_{2,j}^T \mathbf{X}_2]\|_2 = $ LHS. (See Appendix E.)

Note that LHS $\leq \sum_j \|\mathbf{W}_1 - \mathbf{W}_2\|_2 (\lambda \|\tilde{\mathbf{x}}_{u,j} - \mathbf{x}_{2,\phi_2(j)}\|_2 + \|\mathbf{X}_2\|_2)$ and LHS $\leq \sum_j \|\mathbf{W}_1 - \mathbf{W}_2\|_2 ((1-\lambda)\|\mathbf{x}_{1,\phi_1(j)} - \tilde{\mathbf{x}}_{u,j}\|_2 + \|\mathbf{X}_1\|_2)$. (See Appendix E.) By combining the two inequalities above, we have LHS $\leq \sum_j \|\mathbf{W}_1 - \mathbf{W}_2\|_2 \min\{\lambda \|\tilde{\mathbf{x}}_{u,j} - \mathbf{x}_{2,\phi_2(j)}\|_2 + \|\mathbf{X}_2\|_2, (1-\lambda)\|\lambda \mathbf{x}_{1,\phi_1(j)} - \tilde{\mathbf{x}}_{u,j}\|_2 + \|\mathbf{X}_1\|_2\}$. In practice, it is easy to normalize all the tasks in the feature space, which leads to $\|\mathbf{X}_1\|_2 = \|\mathbf{X}_2\|_2$. Therefore, by minimizing the EMD loss $\mathrm{d}_{EMD} = \min\{\min_{\phi_2} \sum_j \|\tilde{\mathbf{x}}_{u,j} - \mathbf{x}_{2,\phi_2(j)}\|_2, \min_{\phi_1} \sum_j \|\tilde{\mathbf{x}}_{u,j} - \mathbf{x}_{1,\phi_1(j)}\|_2\}$, the proposed task up-sampling network identifies from the candidate set $\{\tilde{T}_u\}$ the task $T_u$ that has the minimal distance between $\mathbf{Y}_u$ and $f_{\theta_u}(\mathbf{X}_u)$; in other words, the task-awareness is maximized. $\square$

Previous task augmentation approaches directly mix up two tasks without minimizing the EMD loss, i.e., $\mathbf{y}_{u,j} = (1-\lambda)\mathbf{y}_{1,j} + \lambda \mathbf{y}_{2,j}, \mathbf{x}_{u,j} = (1-\lambda)\mathbf{x}_{1,j} + \lambda \mathbf{x}_{2,j}$. In this case, the task-awareness is unwarranted as we have illustrated in Section 1, provided that $\|\mathbf{Y}_u - f_{\theta_u}(\mathbf{X}_u)\|_2 = \sum_j \|(1-\lambda)\mathbf{y}_{1,j} + \lambda \mathbf{y}_{2,j} - [(1-\lambda)\mathbf{W}_1 + \lambda \mathbf{W}_2][(1-\lambda)\mathbf{x}_{1,j} + \lambda \mathbf{x}_{2,j}]\|_2 = \sum_j \lambda^2(1-\lambda)^2 \|(\mathbf{W_1} - \mathbf{W_2})(\mathbf{x}_{1,j} - \mathbf{x}_{2,j})\|_2$.

# 6  Experiments

To evaluate the effectiveness of ATU, we conduct extensive experiments to answer the following questions: **Q1:** How does ATU perform compared to state-of-the-art task-augmentation-based and regularization meta-learning methods? **Q2:** Whether can the proposed ATU consistently improve performance for different meta-learning methods? **Q3:** What does up-sampled task by ATU looks like? **Q4:** What is the influence of increasing the task number within meta-training data on the performance improvement of ATU? **Benchmarks.** We compared ATU with state-of-the-art task augmentation strategies for meta-learning, including MetaAug [23], MetaMix [40], Meta-Maxup [20], MLTI [43], and regularization methods, including MetaDropout [13], TAML [11], and Meta-Reg [44]for both regression and classification problems. We also consider a variant of ATU which removes the adversarial loss $\mathcal{L}_{adv}$ and trains the task augmentation network only through the EMD loss. We denote this variant by TU. To validate the consistent effect of ATU in improving different meta-learners, we apply ATU and AU on MAML [15], MetaSGD [15] and ANIL [22]. We also consider cross-domain settings where the meta-testing tasks are from different domains.

Table 2: MSE with $\pm 95\%$ confidence intervals on sinusoidal regression.

| Model | 10-shot | 20-shot | 30-shot |
|---|---|---|---|
| DropGrad [32] | $0.91 \pm 0.17$ | $0.62 \pm 0.12$ | $0.55 \pm 0.13$ |
| MetaAug [23] | $0.93 \pm 0.18$ | $0.65 \pm 0.14$ | $0.58 \pm 0.12$ |
| *Meta-Learner: MAML* | | | |
| MAML [9] | $0.93 \pm 0.18$ | $0.65 \pm 0.13$ | $0.58 \pm 0.12$ |
| MetaMix [40] | $0.81 \pm 0.17$ | $0.58 \pm 0.12$ | $0.56 \pm 0.11$ |
| MLTI [43] | $0.92 \pm 0.17$ | $0.65 \pm 0.13$ | $0.62 \pm 0.12$ |
| TU | $0.84 \pm 0.16$ | $0.55 \pm 0.12$ | $0.47 \pm 0.10$ |
| ATU | $\mathbf{0.70 \pm 0.14}$ | $\mathbf{0.47 \pm 0.13}$ | $\mathbf{0.42 \pm 0.11}$ |
| *Meta-Learner: MetaSGD* | | | |
| MetaSGD [15] | $0.70 \pm 0.14$ | $0.49 \pm 0.11$ | $0.42 \pm 0.09$ |
| MetaMix [40] | $0.60 \pm 0.15$ | $0.37 \pm 0.09$ | $0.37 \pm 0.08$ |
| MLTI [43] | $0.66 \pm 0.16$ | $0.51 \pm 0.11$ | $0.44 \pm 0.10$ |
| TU | $0.54 \pm 0.11$ | $0.36 \pm 0.08$ | $0.31 \pm 0.08$ |
| ATU | $\mathbf{0.49 \pm 0.10}$ | $\mathbf{0.34 \pm 0.08}$ | $\mathbf{0.29 \pm 0.08}$ |

Table 3: MSE with $\pm 95\%$ confidence intervals on cross-domain sinusoidal regression.

| Cross-domain | Frequency [0.4,0.8] | Aplitude [5.0,6.0] | Phase $[-\pi,0]$ |
|---|---|---|---|
| *Meta-Learner: MAML* | | | |
| MAML [9] | $1.78 \pm 0.35$ | $3.52 \pm 0.35$ | $3.12 \pm 0.52$ |
| MetaMix [40] | $1.67 \pm 0.30$ | $3.60 \pm 0.28$ | $3.14 \pm 0.54$ |
| MLTI [43] | $1.92 \pm 0.42$ | $3.56 \pm 0.37$ | $3.66 \pm 0.63$ |
| TU | $1.70 \pm 0.34$ | $3.22 \pm 0.31$ | $2.88 \pm 0.48$ |
| ATU | $\mathbf{1.58 \pm 0.35}$ | $\mathbf{2.92 \pm 0.29}$ | $\mathbf{2.58 \pm 0.48}$ |
| *Meta-Learner: MetaSGD* | | | |
| MetaSGD [15] | $2.24 \pm 0.46$ | $2.42 \pm 0.32$ | $2.73 \pm 0.56$ |
| MetaMix [40] | $1.77 \pm 0.35$ | $2.50 \pm 0.27$ | $2.46 \pm 0.48$ |
| MLTI [43] | $1.80 \pm 0.42$ | $2.56 \pm 0.28$ | $2.54 \pm 0.54$ |
| TU | $\mathbf{1.64 \pm 0.38}$ | $2.37 \pm 0.25$ | $\mathbf{2.04 \pm 0.46}$ |
| ATU | $1.71 \pm 0.40$ | $\mathbf{2.19 \pm 0.23}$ | $2.53 \pm 0.62$ |

## 6.1  Regression

**Experimental Setup.** Following [15], we construct the K-shot regression task by sampling from the target sine curve $y(x) = A\sin(\omega x + b)$, where the amplitude $A \in [0.1, 5.0]$, the frequency $\omega \in [0.8, 1.2]$, the phase $b \in [0, \pi]$ and $x$ is sampled from $[-5.0, 5.0]$. In the meta-training phase, each task contains K support and K target (K=10) examples. We adopt mean squared error (MSE) as the loss function. For the base model $f_\theta$, we adopt a small neural network, which consists of an input layer of size 1, 2 hidden layers of size 40 with ReLU and an output layer of size 1. We use one

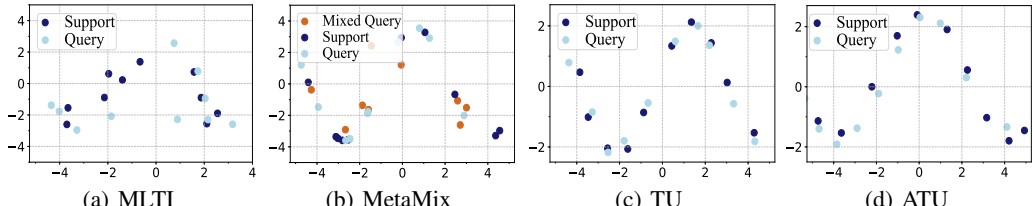

Figure 3: The augmented regression tasks generated by different augmentation-based methods.

gradient update with a fixed step size $\alpha = 0.01$ in inner loop, and use Adam as the outer-loop optimizer following [9, 15]. Moreover, the meta-learner is trained on 240,000 tasks with meta batch-size being 4. In meta-testing stage, we randomly sample 100 sine curves as meta-test tasks, each task containing K support samples and 100 query examples. The data points $x$ in query set are evenly distributed on $[-5.0, 5.0]$. The averaged MSE with 95% confidence intervals upon these 100 sine curves with K=10, 20, 30 are reported in Table 2. We also perform cross-domain experiments by sampling 100 sine curves which have different frequencies, amplitudes or phases from the tasks in meta-training set and report the results in Table 3. More settings about the up-sampling networks are listed in Appendix C.

**Performance.** The results in Table 2 and Table 3 show that ATU consistently outperforms the baseline methods MAML, MetaMix, and MLTI in different K-shot (K$\in \{10, 20, 30\}$) settings and various domain settings. These results validate that the tasks generated by ATU can better approximate the true task distribution and provide more information to the meta-learner than MetaMix and MLTI, thus enabling better generalization of the model. We further verify the superiority of the proposed methods by visualizing the augmented tasks generated by the proposed methods and the baseline methods. The visualization results in Fig. 3 show that the points in the tasks generated by TU and ATU fit the sine curve well, while the points in the tasks generated by MLTI and MetaMix deviate from the sine curve. This indicates that augmented tasks generated by TU and ATU match the true task distribution. It is noteworthy that the support set and query set generated by ATU differ significantly from those generated by TU, which indicates that the task generated by ATU is more difficult. This, together with the results that ATU outperforms TU in most experiments, demonstrates the effectiveness of the adversarial losses in generating informative tasks to improve generalization of the meta-learner.

In Fig. 4, we also visualize the adaptation of meta-learner trained by different task augmentation methods for a 10-shot meta-test regression task. Compared to the MAML trained on original meta-training tasks, the MAML trained on tasks generated by TU fits the ground-truth sinusoid after only one update. And ATU performs even better than TU. This again validates that the augmented tasks generated by TU and ATU are more informative for the meta-learner to learn the meta knowledge from the true task distribution.

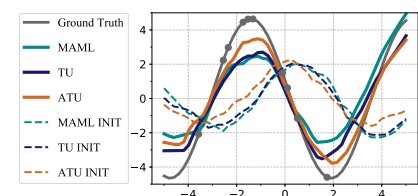

Figure 4: Initialization (dotted) and one-step adaptation (solid) regression curves of MAML, TU and ATU when K=10.

## 6.2 Classification

**Experimental setup.** We follow MLTI to evaluate the performance of task augmentation algorithms for few-shot classification problem with limited number of base classes in meta-training set under non-label-sharing settings. We consider four datasets (base classes number): miniImagenet-S (12), ISIC [18] (4), Dermnet-S (30), and Tabular Murris [5] (57) covering classification tasks on general natural images, medical images, and gene data. Note that the miniImagenet-S and Dermnet-S are constructed by limiting the base classes of miniImangenet [33] and Dermnet [1], respectively. We construct $N$-way $K$-shot tasks, setting $N = 5$ for miniImagenet-S, Derment-S, Tabular Murris and $N = 2$ for ISIC dataset due to its limited number of base classes and setting $K = 1$ or $K = 5$. Recall that TAU relies on extra $K_M$ images to generate augmented tasks in image classification problem. To make the training process more efficient, we set $K_M$ to 3 and the upsampling rate $r$ be 2. More details of these datasets and the settings of the task augmentation networks are listed in Appendix D.

Table 4: Average accuracy under different settings of few-shot classification and various datasets.

| Model | miniImagenet-S | | ISIC | | DermNet-S | | Tabular Murris | |
| --- | --- | --- | --- | --- | --- | --- | --- | --- |
| | 1-shot | 5-shot | 1-shot | 5-shot | 1-shot | 5-shot | 1-shot | 5-shot |
| MAML [9] | 38.27% | 52.14% | 57.59% | 65.24% | 43.47% | 60.56% | 79.08% | 88.55% |
| Meta-Reg [44] | 38.35% | 51.74% | 58.57% | 68.45% | 45.01% | 60.92% | 79.18% | 89.08% |
| TAML [11] | 38.70% | 52.75% | 58.39% | 66.09% | 45.73% | 61.14% | 79.82% | 89.11% |
| Meta-Dropout [13] | 38.32% | 52.53% | 58.40% | 67.32% | 44.30% | 60.86% | 78.18% | 89.25% |
| MetaMix [40] | 39.43% | 54.14% | 60.34% | 69.47% | 46.81% | 63.52% | 81.06% | 89.75% |
| Meta-Maxup [20] | 39.28% | 53.02% | 58.68% | 69.16% | 46.10% | 62.64% | 79.56% | 88.88% |
| MLTI [43] | 41.58% | 55.22% | 61.79% | 70.69% | 48.03% | 64.55% | 81.73% | 91.08% |
| TU | 42.16% | 56.33% | 62.03% | 73.97% | 48.07% | 64.81% | 81.88% | 91.15% |
| ATU | **42.60%** | **56.78%** | **62.84%** | **74.50%** | **48.33%** | **65.16%** | **82.04%** | **91.42%** |

**Performance.** We show the performance on the four datasets in Table 4. On all four datasets, the proposed ATU consistently outperforms the baseline methods, including the augmentation-based methods (i.e. MetaMix, Meta-Maxup and MLTI) and regularization-based methods (Meta-Reg, TAML and Meta-Dropout). And TU achieves the second best performance on all experiments. We also observe that our method achieves a large improvement on the ISIC dataset which consists of only 4 base classes, indicating the effectiveness of our method in limited tasks scenarios. We further evaluate the effectiveness of the proposed ATU on improving the generalization for different backbone meta-learner by conducting experiments under 1-shot setting to compare the performance of MLTI and ATU in improving the performance of the backbone meta-learner MetaSGD and ANIL. The results are presented in Table 5. ATU again consistently outperforms MLTI. All these results validate the superiority of the proposed ATU and TU over the existing baselines in generating informative tasks to improve the performance of different backbone meta-learners. We also evaluate the performance of ATU in cross-domain adaptaion settings. In Table 6, we present the results of the experiment that apply the meta-model trained on miniImageNet-S to Dermnet-S, and vice versa. ATU improves the generalization performance of MAML (the backbone meta-learner in this experiment) by a large margin. This indicates ATU can consistently improve the backbone meta-model's generalization ability under the challenging cross-domain settings.

Table 5: Comparison of compatibility with different backbone meta-learning algorithms on 1-shot classification.

| Method | mini-S | ISIC | Derm-S | TM |
| --- | --- | --- | --- | --- |
| MetaSGD [15] | 37.88% | 58.79% | 42.07% | 81.55% |
| MetaSGD+MLTI | 39.58% | 61.57% | 45.49% | 83.31% |
| MetaSGD+ATU | **40.52%** | **62.84%** | **46.78%** | **83.84%** |
| ANIL [22] | 38.02% | 59.48% | 44.58% | 75.67% |
| ANIL+MLTI | 39.15% | 61.78% | 46.79% | 77.11% |
| ANIL+ATU | **39.27%** | **62.12%** | **47.03%** | **77.23%** |

Table 6: Cross-domain adaptation experiments between mini-S and Dermnet-S. A→B denotes that the backbone meta-model is meta-trained on A and meta-tested on B.

| Model | mini-S→ Derm-S | | Derm-S→ mini-S | |
| --- | --- | --- | --- | --- |
| | 1-shot | 5-shot | 1-shot | 5-shot |
| MAML [9] | 34.46% | 50.36% | 28.78% | 41.29% |
| MAML+ATU | **36.86%** | **51.98%** | **30.68%** | **46.72%** |
| MetaSGD [15] | 31.07% | 49.07% | 28.17% | 41.83% |
| MetaSGD+ATU | **37.75%** | **54.60%** | **30.78%** | **44.01%** |

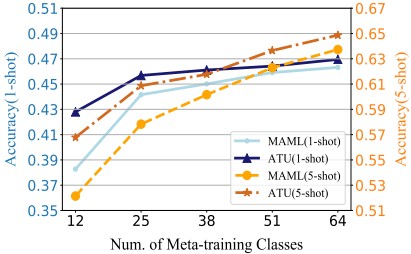

Figure 5: The averaged accuracy on the miniImagenet dataset with different number of tasks.

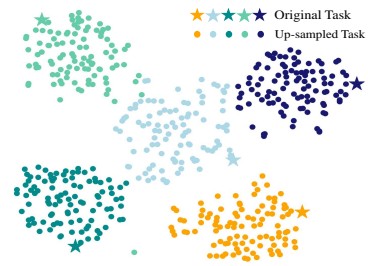

Figure 6: T-SNE visualization of original and up-sampled tasks on 1-shot miniImagenet-S setting.

**Effect of the number of meta-training tasks.** We conduct experiments to analyze the change in the performance improvement of ATU over MAML with the number of meta-training tasks in 1-shot and 5-shot settings. The results presented in Fig. 5 show that ATU significantly improves the performance of MAML by about 4.5%-5% when the number of base classes is 12, while the improvement decreases with the number of base classes increasing on both 1-shot and 5-shot settings. When the number of base classes increases, the number of training tasks increases rapidly and the empirical task distribution constructed from the meta-training tasks becomes closer and closer to the true latent task distribution. Therefore, the extra information provided by tasks generated by ATU becomes less. However, even if all available base classes are used in the meta-training (i.e., 64 meta-training classes), our proposed ATU could still help to improve the performance of MAML.

**Visualization of the generated tasks.** We visualize the up-sampled tasks by t-SNE to evaluate their generation quality for MAML under the 1-shot miniImagenet-S setting. Concretely, we up-sample 100 tasks for 5 original tasks via ATU by using different perturbations for each task. In order to visualize the relationship between generated tasks and original tasks using t-SNE, we represent each task by concatenating the vector of the support and query sets. The results presented in Fig. 6 show that the up-sampled tasks stay near the original tasks, which means they are matching the true task distribution. This indicates the generated tasks are task-aware. Besides, we can observe that the augmented tasks are diverse and cover a substantial portion of the original tasks. This demonstrates the task imaginary property of the augmented tasks. These two observations suggest that the proposed ATU is a qualified task augmentation algorithm.

**Effect of the extra** $d_{EMD}(\hat{D}_i^s, \hat{D}_i^q)$ **in regression tasks.** As presented in Section 4.1, we propose to apply an extra EMD loss $d_{EMD}(\hat{D}_i^s, \hat{D}_i^q)$ on the support and query set for each generated task to encourage the points in the generated support set and the query set to follow the same sine curve. In Fig. 3, we have visualized the tasks generated by the Task Up-sampling Network trained with the extra EMD loss. Here we provide additional visualization result for tasks generated by the Task Up-sampling Network trained without the extra EMD loss in Fig. 7. It can be seen that the support set and the query set are not on the same sinusoid, indicating that the generated tasks need additional supervision to avoid them being too difficult or not even valid tasks.

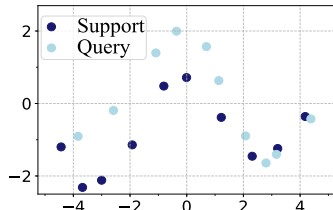

Figure 7: Visualization of the up-sampled task generated by ATU when $\eta_3 = 0$ in Eq. (6).

## 7 Conclusion and Limitation

In this paper, we propose the first task-level up-sampling network that learns to generate tasks that simultaneously meet the qualifications of being task-aware, task-imaginary, and model-adaptive. The proposed Adversarial Task Up-sampling (ATU) takes a set of tasks as input and learn to up-sample tasks complying with the true task distribution while being informative to improve the generalization of the meta-learner. We theoretically justify that ATU promotes task-awareness and empirically verify that ATU improves the generalization of various backbone meta-learner for both regression and classification tasks on five datasets. **Limitations.** Our theoretical results are obtained under some strong assumptions, but all the experiments and visualization outcomes validate our method's effectiveness in real settings.

## 8 Acknowledgement

This work is sponsored by the Tencent AI Lab Gift Fund (Project 9229073) and CityU Strategic Interdisciplinary Research Grant (Project 7020064).

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
