# A  Pseudo-codes

We present the pseudo-codes for the task upsampling network $g(\cdot)$ (Algorithm 1), and the meta-training algorithm for the regression task (Algorithm 2) and the classification task (Algorithm 3) in 1-step MAML with ATU. For regression task, we randomly sample a batch of tasks as the ground-truth task set $\mathcal{T}_g$ and construct the task patch by down-sampling (FPS sampling). For classification tasks, we construct $(K^s + K^q)$ tasks in one shot to obtain a task patch from a $K^s$-shot classification task with $K^q$ query samples. We assume the local task distribution to be smooth, and construct the ground-truth tasks by perform mixup for each image in each task with a nearest image in $K_M$ images. We name the set of $K_M$ images by *memory bank* and denote it by $\mathcal{I}_M$. The $K_M$ images are randomly sampled from classes different from those in the input tasks. It worth note that ATU is only applied in the meta-training phase and, therefore, the meta-testing phase remains the same as the original 1-step MAML. It is direct to extend 1-step MAML to multi-step MAML and extend MAML to ANIL, Meta-SGD without modifying ATU.

---

**Algorithm 1** Task Up-sampling Network $g(\cdot)$

---

**Require:**  up-sampling ratio $r = r_c \times r_d$ for the coarse generator and decoder, respectively
 1: **Input:** a task patch $\mathcal{T}_p = \{T_i\}_{i=1}^{N_p}$
 2: Extract the set feature for the input task patch $h_s = g_s(\mathcal{T}_p)$
 3: Generate a set of coarse tasks $\mathcal{T}_c = g_c(\mathcal{T}_p)$ with set size $r_c N_p$
 4: Sample a set of perturbations $\mathcal{Z}$ in size $r_d$
 5: Generate the up-sampled task set $\mathcal{T}_{up} = g_d(\mathcal{T}_c, \mathcal{Z}, h_s)$
 6: **Output:** a set of up-sampled tasks $\mathcal{T}_{up}$

---

**Algorithm 2** Meta-training of 1-step MAML with ATU for regression tasks

---

**Require:**  distribution over meta-training tasks $p(\mathcal{T})$; inner-loop and outer-loop learning rates $\alpha,\beta$; hyperparameters $\eta_1, \eta_2, \eta_3$ in Eq. (4) and Eq. (6); batch size of tasks $B$; task patch size $N_p$; up-sampling ratio $r = r_c \times r_d$ for the task up-sampling network
 1: Randomly initialize the parameters $\theta_0$ of the meta-model.
 2: **while** not converge **do**
 3:     Randomly sample a batch of tasks as $\mathcal{T}_g$ with batch size $r N_p$
 4:     Perform down-sampling (FPS sampling) on $\mathcal{T}_g$ to construct the local task patch $\mathcal{T}_p$
 5:     Generate the augmented task set through our Task Up-sampling Network as $\mathcal{T}_{up} = g(\mathcal{T}_p)$
 6:     Randomly split the up-sampled task set $\mathcal{T}_{up}$ into $n$ batches $\{\mathcal{T}_{batch}\}$, each with $B$ tasks (i.e., $n = |\mathcal{T}_{up}|/B$)
 7:     **for** each task batch $\mathcal{T}_{batch}$ in $\mathcal{T}_{up}$ **do**
 8:         **for** each task $T_i \in \mathcal{T}_{batch}$ **do**
 9:             Perform inner-loop update of MAML as $\phi_i = \theta_0 - \alpha \nabla_{\theta_0} \mathcal{L}(f_{\theta_0}, D_i^s)$
10:         **end for**
11:         Calculate $\mathcal{L}(f_{\phi_i}, D_i^q)$ and $\mathcal{L}_{adv}(\theta_0, D_i^s, D_i^q)$
12:         Update the meta-model parameter $\theta_0$ as $\theta_0 \leftarrow \theta_0 - \beta \frac{1}{B} \sum_{i=1}^{n} \nabla_{\theta_0} \mathcal{L}(f_{\phi_i}, D_i^q)$
13:     **end for**
14:     Calculate the objective function in Eq. (6) and perform backpropagation to update the Task Up-sampling Network
15: **end while**

---

# B  Network Architecture of the Task Up-sampling Network

In this section, we provide the network architectures for the Task Up-sampling Network for both regression and classification tasks. As shown in Fig. 8, the set encoder $g_s(\cdot)$ of the Task Up-Sampling Network for regression tasks consists of 2 convolution layers followed by a max-pooling layer to extract the permutation-invariant feature for the input task patch. The dimension of the set feature is 1024. The coarse task generator $g_c(\cdot)$ consists of a set encoder to extract the set feature for the input patch, followed by 3 linear layers to generate coarse tasks from the set feature. The set encoder in coarse generator is the same as $g_s(\cdot)$. The output of the last layers is reshaped into $r_c N_p$ coarse tasks. By concatenating the coarse task and a $r_d$-dimension noise vector, we obtain the input of the

**Algorithm 3** Meta-training of 1-step MAML with ATU for classification tasks (N-way $K^s$-shot)

**Require:** distribution over meta-training tasks $p(\mathcal{T})$; inner-loop and outer-loop learning rates $\alpha, \beta$; hyperparameters $\eta_1, \eta_2$ in Eq. (4); batch size of tasks $B$; task patch size $N_p$; Beta distribution $Beta(\delta_1, \delta_2)$; up-sampling ratio $r$ ($r = r_c \times r_d, r_c = 1$)
1: Randomly initialize the parameters $\theta_0$ of the meta model
2: **while** not converge **do**
3:     Randomly sample a batch of tasks $\mathcal{T}_{batch}$ with $B$ tasks.
4:     **for** each task $T_i \in \mathcal{T}_{batch}$ **do**
5:         Reshape $T_i$ as the task patch $\mathcal{T}_p$
6:         Randomly sample extra $K_M$ images which consists of images not belong to any class in $T_i$
7:         Construct the $\mathcal{T}_g = (\hat{C}_0, ..., \hat{C}_N)$ : Sample $\lambda \sim Beta(\delta_1, \delta_2)$. For the image in each class $C_j$ in $T_i$, generate a new image as $\hat{C}_j = \lambda \times C_j + (1 - \lambda) \times X_j$, where $X_j$ is the nearest image (measured by euclidean distance) to the image in the class $C_j$.
8:         Generate up-sampling task set through Task Up-sampling Network as $\mathcal{T}_{up} = g(\mathcal{T}_p)$
9:         Randomly sample one task $\hat{T}_i$ from $\mathcal{T}_{up}$
10:        Perform inner-loop update of MAML as $\phi_i = \theta_0 - \alpha \nabla_{\theta_0} \mathcal{L}(f_{\theta_0}, \hat{D}_i^s)$
11:     **end for**
12:     Calculate $\mathcal{L}(f_{\phi_i}, D_i^q)$ and $\mathcal{L}_{adv}(\theta_0, D_i^s, D_i^q)$
13:     Update the meta model parameter $\theta_0$ as $\theta_0 \leftarrow \theta_0 - \beta \frac{1}{B} \sum_{i=1}^n \nabla_{\theta_0} \mathcal{L}(f_{\phi_i}, \hat{D}_i^q)$
14:     Calculate the objective function in Eq. (5) and perform backpropagation to update the Task Up-sampling Network
15: **end while**

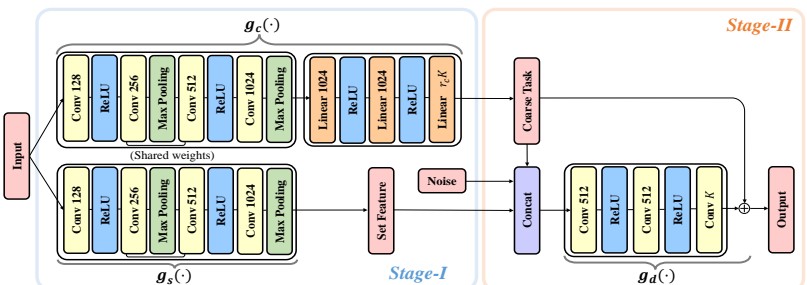

Figure 8: The up-sampling network of the regression task.

decoder $g_d(\cdot)$. The decoder consists of 3 convolution layers. We then use the output of the last layer as residual added to the coarse tasks to obtain the up-sampling tasks.

The network structure of Task Up-sampling Network for classification tasks is presented in Fig. 9. We directly use the input task patch as the coarse tasks and, therefore, the coarse task generator $g_c(\cdot)$ is an identity function. The set encoder $g_s(\cdot)$ consists of 2 convolution layers, each followed by a Batch Norm layer. We use the $K_M$ images in the memory bank as the perturbation, concatenating with a $(N_p \times K_M)$-dimension noise vector to obtain the input to the decoder. The decoder consists of a attention module and a mapping module. The attention module is constructed by 3 convolution layers, followed by 3 linear layers. The attention block generates the attention scores. The mapping module, which consists of 3 convolution layers with xxx filters, maps the $K_M$ perturbation to $K_M$ residual features. We perform weighted sum of the $K_M$ residual features with the attention scores to generate $r$ final residual features and add them to the coarse tasks to obtain the up-sampling 1-shot tasks. We then construct $r$ augmented tasks by stacking $K^s + K^q$ 1-shot tasks.

## C   Setups and Additional Experiment Results for Regression Tasks

### C.1   Setups of Hyperparameter

The hyperparameters of the ATU are listed in Table 8.

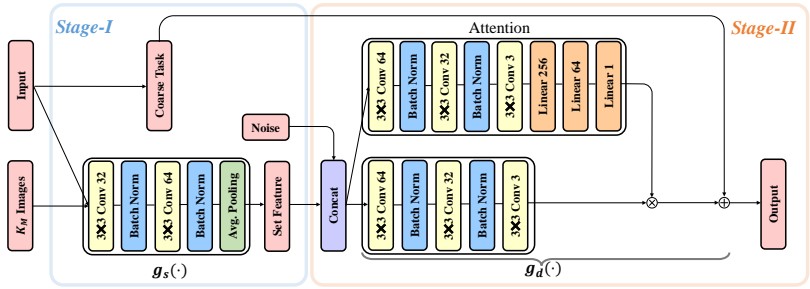

Figure 9: The up-sampling network of the classification task.

## C.2 Effect of Augmentation Ratio

In the regression task, we assume the combination of the augmented and original tasks will better approximate the real task distribution and, therefore, update the meta-model not only with the augmented tasks generated by the Task Up-sampled Network, but also with the original meta-training tasks. We define the augmentation ratio as the proportion of augmented tasks among all the tasks. Note that the experiment results shown in Table 2 are obtained by setting the augmented ratio as 0.2. We present the results with other ratios in Table 7. It can be observed that the performance with positive ratio is better than the performance with ratio= 0, which means the meta-model is trained without augmented tasks. These results indicates that the augmented tasks are more informative than the original tasks in training a better meta-model.

Table 7: Ablation study on the augmentation ratio of the sine regression task.

| Augmentation ratio | TU performance (10-shot) |
|---|---|
| 0 | $0.93 \pm 0.18$ |
| 0.2 | $0.84 \pm 0.16$ |
| 0.4 | $0.89 \pm 0.17$ |
| 0.6 | $0.91 \pm 0.18$ |

Table 8: Hyperparameters of the sine regression task in Table 2.

| Hyperparameters | ATU |
|---|---|
| maximum training iterations | 3750 |
| up-sampling ratio r $(r_c, r_d)$ | 8 (2, 4) |
| loss weights $(\eta_1, \eta_2, \eta_3)$ | $(8e^{-3}, 4e^{-3}, 3e^{-1})$ |
| size of $\mathcal{T}_g$ | 64 |

# D Setups and Additional Experiment Results for Classification Tasks

## D.1 Introduction and Hyperparameters of the Four Datasets

We provide the detailed information of the datasets and hyperparameters of the classification tasks for obtaining the results in Tabel 4 in this section. We construct the 4 datasets following the settings in MLTI [43].

**miniImagenet-S.** Compared with miniImagenet, miniImagenet-S has fewer meta-training classes so as to limit the task number. The specific meta-training classes of miniImagenet-S include:

*n03017168, n07697537, n02108915, n02113712, n02120079, n04509417,n02089867, n03888605, n04258138, n03347037, n02606052, n06794110*

We use four convolutional blocks and a classifier as the base learner [43, 9], and each convolutional block contains a convolutional layer, a batch normalization layer and a ReLU activation layer. In order to analyze the effect of the number of meta-training tasks, we add more classes for meta-training according to the following sequence:

*n03476684, n02966193, n13133613, n03337140, n03220513, n03908618,n01532829, n04067472, n02074367, n03400231, n02108089, n01910747, n02747177, n02795169, n04389033, n04435653, n02111277, n02108551,n04443257, n02101006, n02823428, n03047690, n04275548, n04604644, n02091831, n01843383, n02165456, n03676483, n04243546, n03527444, n01770081, n02687172, n09246464, n03998194, n02105505, n01749939, n04251144, n07584110, n07747607, n04612504, n01558993, n03062245, n04296562, n04596742, n03838899, n02457408, n13054560, n03924679, n03854065, n01704323, n04515003, n03207743*

**ISIC.** ISIC skin dataset [18] was provided by ISIC2018 Challenge, in which 7 disease classes and 10015 dermoscopic images are included. Following [43, 14], *Nevus, Malanoma, Benign Keratoses, Basal Cell Carcinoma*, the four categories with the largest number of images, are as meta-training classes; the rest *Dermatofibroma, Pigmented Bowen's, Benign Keratoses* are as meta-testing classes. We re-scale the size of each medical image to $84 \times 84 \times 3$ and adopt the same 4-layer convolutional as the base model like miniImagenet-S.

**DermNet-S.** Dermnet-S are part of the public Dermnet Skin Disease Altas, in which 625 different fine-grained categories are included. Dermnet-S chooses the top-30 classes for meta-training. The concrete meta-training classes and meta-testing classes are:

- **Meta-training classes:** *Seborrheic Keratoses Ruff, Herpes Zoster, Atopic Dermatitis Adult Phase, Psoriasis Chronic Plaque, Eczema Hand, Seborrheic Dermatitis, Keratoacanthoma, Lichen Planus, Epidermal Cyst, Eczema Nummular, Tinea (Ringworm) Versicolor, Tinea (Ringworm) Body, Lichen Simplex Chronicus, Scabies, Psoriasis Palms Soles, Malignant Melanoma, Candidiasis large Skin Folds, Pityriasis Rosea, Granuloma Annulare, Erythema Multiforme, Seborrheic Keratosis Irritated, Stasis Dermatitis and Ulcers, Distal Subungual Onychomycosis, Allergic Contact Dermatitis, Psoriasis, Molluscum Contagiosum, Acne Cystic, Perioral Dermatitis, Vasculitis, Eczema Fingertips.*

- **Meta-testing classes:** *Warts, Ichthyosis Sex Linked, Atypical Nevi, Venous Lake, Erythema Nodosum, Granulation Tissue, Basal Cell Carcinoma Face, Acne Closed Comedo, Scleroderma, Crest Syndrome, Ichthyosis Other Forms, Psoriasis Inversus, Kaposi Sarcoma, Trauma, Polymorphous Light Eruption, Dermagraphism, Lichen Sclerosis Vulva, Pseudomonas, Cutaneous Larva Migrans, Psoriasis Nails, Corns, Lichen Sclerosus Penis, Staphylococcal Folliculitis, Chilblains Perniosis, Psoriasis Erythrodermic, Squamous Cell Carcinoma Ear, Basal Cell Carcinoma Ear, Ichthyosis Dominant, Erythema Infectiosum, Actinic Keratosis Hand, Basal Cell Carcinoma Lid, Amyloidosis, Spiders, Erosio Interdigitalis Blastomycetica, Scarlet Fever, Pompholyx, Melasma, Eczema Trunk Generalized, Metastasis, Warts Cryotherapy, Nevus Spilus, Basal Cell Carcinoma Lip, Enterovirus, Pseudomonas Cellulitis, Benign Familial Chronic Pemphigus, Pressure Urticaria, Halo Nevus, Pityriasis Alba, Pemphigus Foliaceous, Cherry Angioma, Chapped Fissured Feet, Herpes Buttocks, Ridging Beading.*

**Tabular Murris.** The Tabular Murris is a gene dataset (i.e., 2866-dim features) including 105,960 cells of 124 cell types extracted from 23 organs. Following [43, 5], the concrete training/validation/testing split is:

- **Meta-training classes:** *BAT, MAT,Limb Muscle, Trachea, Heart, Spleen, GAT, SCAT, Mammary Gland, Liver, Kidney, Bladder, Brain Myeloid, Brain Non-Myeloid, Diaphragm.*

- **Meta-validation classes:** *Skin, Lung, Thymus, Aorta*

- **Meta-testing organs:** *Large Intestine, Marrow, Pancreas, Tongue*

Unlike the base model for the other 3 datasets, we use two fully connected blocks and a linear layer as the backbone network, where each fully connected block includes a linear layer, a batch normalization layer, a ReLu activation layer, and a dropout layer. The dropout ratio and the feature channels of the linear layer are set 0.2, 64, which is the same as the settings of [43, 5].

For a fair comparison with MLTI, we adopt the same MetaMix strategy as MLTI to augment the query set for the four datasets. We set the augmentation ratio as 1 in classification tasks and do not use the original sampled tasks in meta-training because we empirically find that ATU obtains better performance with higher augmentation ratio. The other settings are the same with MLTI. More details about the hyperparameters are listed in Table 9.

Table 9: Hyperparameters of Tabel 4

| Hyperparameters(ATU) | miniImagenet-S | ISIC | Dermnet-S | Tabular Murris |
|---|---|---|---|---|
| inner-loop learning rate | 0.01 | 0.01 | 0.01 | 0.01 |
| outer-loop learning rate | 0.001 | 0.001 | 0.001 | 0.001 |
| $Beta(\delta_1, \delta_2)$ | (3,5) | (2,2) | (2,2) | (2,2) |
| Number of steps in inner loop | 5 | 5 | 5 | 5 |
| batch size | 4 | 4 | 4 | 4 |
| query size in meta-training tasks | 15 | 15 | 15 | 15 |
| maximum training iterations | 50,000 | 50,000 | 50,000 | 50,000 |
| adversarial loss weights $\eta$ ($\eta_1 = \eta_2$) | 3 | 3 | 0.5 | 0.5 |
| up-sampling ratio r | 2 | 2 | 2 | 2 |

Table 10: Ablation study on the memory bank size $K_M$ in the classification task.

| $K_M$ | TU (mini-S 1-shot) |
|---|---|
| 3 | $42.16 \pm 0.73\%$ |
| 5 | $42.20 \pm 0.76\%$ |
| 7 | $42.28 \pm 0.72\%$ |

Table 11: Sensitivity analysis of the adversarial loss weights on the classification task.

| Adversarial weights | ATU (mini-S 1-shot) |
|---|---|
| $\eta_1 = \eta_2 = 1$ | $42.38 \pm 0.82\%$ |
| $\eta_1 = \eta_2 = 3$ | $42.60 \pm 0.84\%$ |
| $\eta_1 = \eta_2 = 5$ | $41.67 \pm 0.79\%$ |

## D.2  Ablation Study

**Effect of $K_M$.** As shown in Table 10, the classification performance increases as the memory bank size $K_M$ increases. But the performance gain is not very significant for a large $K_M$. Considering the training efficiency, we set $K_M = 3$ in all classification experiments.

Table 12: The averaged accuracy with 95% confidence intervals of various interpolation task augmentation methods and our task up-sampling method on miniImagenet-s (5-shot).

| Task generation method | miniImagenet-S (5-shot) |
|---|---|
| Naive Baseline[1] | $53.49 \pm 0.74\%$ |
| Naive Baseline[2] | $50.25 \pm 0.71\%$ |
| Naive Baseline[3] | $53.91 \pm 0.78\%$ |
| TU | $56.33 \pm 0.79\%$ |

**Effect of $\eta_1, \eta_2$.** We choose different ($\eta_1, \eta_2$) to explore the sensitivity of model performance to adversarial loss weights. It can be observed from the results in Table 11 that the adversarial loss weights have a large influence on the performance of the model and it achieves the best performance when setting $\eta_1 = \eta_2 = 3$.

**Effect of Augmented Task Generation Strategies.** Due to the high complexity of the classification tasks' distribution, we assume its latent task distribution is smooth and construct the ground-truth task manifold $\mathcal{T}_g$ via mixing all image in each class of $T_i$ with its corresponding nearest image in the memory bank (the sampled $K_M$ images) (see Algorithm 3). Under this assumption, one naive method is to generate augmented tasks in the same way as the generation of ground-truth tasks where we can directly mix the images in the tasks with the $K_M$ images in the memory bank. To verify the effectiveness and necessity of training a Task Up-sampling Network, we compare TU with 3 naive methods in Table 12: (1) Naive Baseline[1]:for each image of task $T_i$, we randomly choose one image in the memory bank to mix; (2) Naive Baseline[2]: for all images in a class of task $T_i$, we randomly choose one image in the memory bank to mix; (3) Naive Baseline[3]: for all images in a class of task $T_i$, we choose the nearest image in the memory bank to mix. The Naive Baseline[3] is the method that we used to construct $\mathcal{T}_g$. The results in Table 12 show that TU outperforms the other 3 baselines by a large margin. TU outperforms Naive Baseline[3] because the tasks generated by TU match the local task distribution better than those generated by just mixing with the images in the memory bank.

Moreover, the tasks generated by TU is more diverse and informative. Taking this into consideration, we set the augmentation ratio to be 1 and do not use the original tasks in the meta-training.

### D.3 Visualization of the generated Classification tasks of $\mathcal{T}_p$.

We visualize part of the images in a generated classification task in Fig. 10. The three images in the top row are the selected extra $K_M$ images and $\hat{T}_1, \hat{T}_2, \hat{T}_3$ are three 5-way 1-shot tasks generated by ATU.

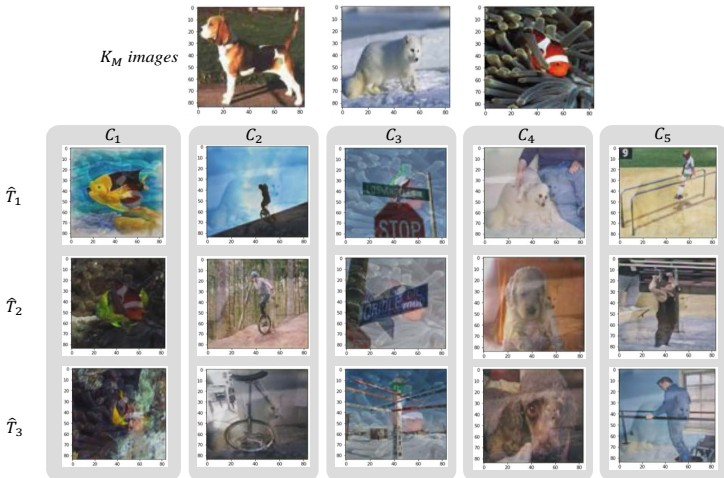

Figure 10: Visualization of part up-sampled classification tasks (i.e., $\mathcal{T}_{up}$) generated by ATU.

Table 13: Complete results of Table 6 with 95% confidence interval under the cross-domain setting.

| Model | miniImagenet-S→ DermNet-S | | DermNet-S→ miniImagenet-S | |
| | 1-shot | 5-shot | 1-shot | 5-shot |
| --- | --- | --- | --- | --- |
| MAML [9] | $34.46 \pm 0.63\%$ | $50.36 \pm 0.64\%$ | $28.78 \pm 0.55\%$ | $41.29 \pm 0.64\%$ |
| MAML+ATU | $\mathbf{36.86 \pm 0.64\%}$ | $\mathbf{51.98 \pm 0.62\%}$ | $\mathbf{30.68 \pm 0.68\%}$ | $\mathbf{46.72 \pm 0.73\%}$ |
| MetaSGD [15] | $31.07 \pm 0.57\%$ | $49.07 \pm 0.59\%$ | $28.17 \pm 0.53\%$ | $41.83 \pm 0.67\%$ |
| MetaSGD+ATU | $\mathbf{37.75 \pm 0.65\%}$ | $\mathbf{54.60 \pm 0.58\%}$ | $\mathbf{30.78 \pm 0.58\%}$ | $\mathbf{44.01 \pm 0.68\%}$ |

### D.4 Complete Results with Confidence Interval

We list the complete results with 95% confidence interval in Table 14,15,13, which are corresponding to the Tabel 4, 5, 6 in Section 6.2.

## E Proof of Property 1.

**Property 1** (Task-awareness Maximization). Consider $N_u = 2$, $g(\theta_1, \theta_2) = (1 - \lambda)\theta_1 + \lambda\theta_2$, $f_{\theta_1}(\cdot) = \mathbf{W}_1$, and $f_{\theta_2}(\cdot) = \mathbf{W}_2$. The proposed ATU algorithm that pursues an up-sampled task $T_u = \{\mathbf{X}_u, \mathbf{Y}_u\}$ via minimizing the EMD loss between $T_1$ and $T_2$ maximizes the task-awareness, i.e., minimizing the distance between $\mathbf{Y}_u$ and $f_{\theta_u}(\mathbf{X}_u)$.

*Proof.* According to the definition of EMD (Eq. (2)), it solves: $\phi^* = \arg\min_{\phi \in \mathbf{\Phi}} \sum_j \|\mathbf{x}_{1,j} - \mathbf{x}_{2,\phi(j)}\|_2$, where $\mathbf{\Phi} = \{\{1, \cdots, n\} \mapsto \{1, \cdots, n\}\}$ denotes the set containing all possible bijective assignments, each of which gives one-to-one correspondence between $T_1$ and $T_2$. Based on the optimal assignments $\phi^*$, the EMD is known to be defined as $d_{EMD} = \frac{1}{n} \sum_j \|\mathbf{x}_{1,j} - \mathbf{x}_{2,\phi^*(j)}\|_2$.

In light of the difficulty in mathematically formulating a possible up-sampled task $\tilde{T}_u$ that lies in the local manifold of $\{T_1, T_2\}$, we reasonably assume a simplified way of characterizing an

Table 14: Complete classification results of Table 4 with 95% confidence interval.

| Setting | Model | miniImagenet-S | ISIC | DermNet-S | Tabular Murris |
|---|---|---|---|---|---|
| 1-shot | MAML [9] | $38.27 \pm 0.74\%$ | $57.59 \pm 0.79\%$ | $43.47 \pm 0.83\%$ | $79.08 \pm 0.91\%$ |
| | Meta-Reg [44] | $38.35 \pm 0.76\%$ | $58.57 \pm 0.94\%$ | $45.01 \pm 0.83\%$ | $79.18 \pm 0.87\%$ |
| | TAML [11] | $38.70 \pm 0.77\%$ | $58.39 \pm 1.00\%$ | $45.73 \pm 0.84\%$ | $79.82 \pm 0.87\%$ |
| | Meta-Dropout [13] | $38.32 \pm 0.75\%$ | $58.40 \pm 1.02\%$ | $44.30 \pm 0.84\%$ | $78.18 \pm 0.93\%$ |
| | MetaMix [40] | $39.43 \pm 0.77\%$ | $60.34 \pm 1.03\%$ | $46.81 \pm 0.81\%$ | $81.06 \pm 0.86\%$ |
| | Meta-Maxup [20] | $39.28 \pm 0.77\%$ | $58.68 \pm 0.86\%$ | $46.10 \pm 0.82\%$ | $79.56 \pm 0.89\%$ |
| | MLTI [43] | $41.58 \pm 0.72\%$ | $61.79 \pm 1.00\%$ | $48.03 \pm 0.79\%$ | $81.73 \pm 0.89\%$ |
| | TU | $42.16 \pm 0.76\%$ | $62.03 \pm 0.95\%$ | $48.07 \pm 0.83\%$ | $81.88 \pm 0.90\%$ |
| | ATU | $\mathbf{42.60 \pm 0.77\%}$ | $\mathbf{62.84 \pm 0.98\%}$ | $\mathbf{48.33 \pm 0.81\%}$ | $\mathbf{82.04 \pm 0.94\%}$ |
| 5-shot | MAML [9] | $52.14 \pm 0.65\%$ | $65.24 \pm 0.77\%$ | $60.56 \pm 0.74\%$ | $88.55 \pm 0.60\%$ |
| | Meta-Reg [44] | $51.74 \pm 0.68\%$ | $68.45 \pm 0.81\%$ | $60.92 \pm 0.69\%$ | $89.08 \pm 0.61\%$ |
| | TAML [11] | $52.75 \pm 0.70\%$ | $66.09 \pm 0.71\%$ | $61.14 \pm 0.72\%$ | $89.11 \pm 0.59\%$ |
| | Meta-Dropout [13] | $52.53 \pm 0.69\%$ | $67.32 \pm 0.92\%$ | $60.86 \pm 0.73\%$ | $89.25 \pm 0.59\%$ |
| | MetaMix [40] | $54.14 \pm 0.73\%$ | $69.47 \pm 0.60\%$ | $63.52 \pm 0.73\%$ | $89.75 \pm 0.58\%$ |
| | Meta-Maxup [20] | $53.02 \pm 0.72\%$ | $69.16 \pm 0.61\%$ | $62.64 \pm 0.72\%$ | $88.88 \pm 0.57\%$ |
| | MLTI [43] | $55.22 \pm 0.76\%$ | $70.69 \pm 0.68\%$ | $64.55 \pm 0.74\%$ | $91.08 \pm 0.54\%$ |
| | TU | $56.33 \pm 0.69\%$ | $73.97 \pm 0.70\%$ | $64.81 \pm 0.72\%$ | $91.15 \pm 0.60\%$ |
| | ATU | $\mathbf{56.78 \pm 0.73\%}$ | $\mathbf{74.50 \pm 0.90\%}$ | $\mathbf{65.16 \pm 0.75\%}$ | $\mathbf{91.42 \pm 0.61\%}$ |

Table 15: Complete results of Table 5 with 95% confidence interval under different backbones.

| Method | miniImagenet-S | ISIC | DermNet-S | Tabular Muris |
|---|---|---|---|---|
| MetaSGD [15] | $37.88 \pm 0.74\%$ | $58.79 \pm 0.82\%$ | $42.07 \pm 0.83\%$ | $81.55 \pm 0.91\%$ |
| MetaSGD+MLTI | $39.58 \pm 0.76\%$ | $61.57 \pm 1.10\%$ | $45.49 \pm 0.83\%$ | $83.31 \pm 0.87\%$ |
| MetaSGD+ATU | $\mathbf{40.52 \pm 0.78\%}$ | $\mathbf{62.84 \pm 1.01\%}$ | $\mathbf{46.78 \pm 0.84\%}$ | $\mathbf{83.84 \pm 0.90\%}$ |
| ANIL [22] | $38.02 \pm 0.75\%$ | $59.48 \pm 1.00\%$ | $44.58 \pm 0.85\%$ | $75.67 \pm 0.99\%$ |
| ANIL+MLTI | $39.15 \pm 0.73\%$ | $61.78 \pm 1.24\%$ | $46.79 \pm 0.77\%$ | $77.11 \pm 1.00\%$ |
| ANIL+ATU | $\mathbf{39.27 \pm 0.76\%}$ | $\mathbf{62.12 \pm 0.98\%}$ | $\mathbf{47.03 \pm 0.85\%}$ | $\mathbf{77.23 \pm 0.99\%}$ |

up-sampled task $\tilde{T}_u$ to be $\tilde{\mathbf{y}}_{u,j} = \boldsymbol{\alpha}_{1,j}^T \mathbf{Y}_1 + \boldsymbol{\alpha}_{2,j}^T \mathbf{Y}_2$, $\tilde{\mathbf{x}}_{u,j} = \boldsymbol{\alpha}_{1,j}^T \mathbf{X}_1 + \boldsymbol{\alpha}_{2,j}^T \mathbf{X}_2$, $\forall j$, where each sample is a convex combination of samples from both $T_1$ from $T_2$. The combination coefficients $\boldsymbol{\alpha}_{1,j}^T, \boldsymbol{\alpha}_{2,j}^T \in \mathbb{R}^{(K^s + K^q) \times 1}$, $\sum_k^{K^s + K^q} \alpha_{1,jk} = 1$, and $\sum_k^{K^s + K^q} \alpha_{2,jk} = 1$. Different combination coefficients lead to a set of up-sampled task candidates $\{\tilde{T}_u\}$. We evaluate the task-awareness property of each candidate $\tilde{T}_u$, i.e., the distance between $\tilde{\mathbf{Y}}_u$ and $f_{\theta_u}(\tilde{\mathbf{X}}_u)$, to be

$$\|\tilde{\mathbf{Y}}_u - f_{\theta_u}(\tilde{\mathbf{X}}_u)\|_2 = \sum_j \|\tilde{\mathbf{y}}_{u,j} - f_{\theta_u}(\tilde{\mathbf{x}}_{u,j})\|_2$$

$$= \sum_j \|\boldsymbol{\alpha}_{1,j}^T \mathbf{Y}_1 + \boldsymbol{\alpha}_{2,j}^T \mathbf{Y}_2 - [(1-\lambda)\mathbf{W}_1 + \lambda \mathbf{W}_2][\boldsymbol{\alpha}_{1,j}^T \mathbf{X}_1 + \boldsymbol{\alpha}_{2,j}^T \mathbf{X}_2]\|_2$$

$$= \sum_j \|\boldsymbol{\alpha}_{1,j}^T \mathbf{X}_1 \mathbf{W}_1 + \boldsymbol{\alpha}_{2,j}^T \mathbf{X}_2 \mathbf{W}_2 - [(1-\lambda)\mathbf{W}_1 + \lambda \mathbf{W}_2][\boldsymbol{\alpha}_{1,j}^T \mathbf{X}_1 + \boldsymbol{\alpha}_{2,j}^T \mathbf{X}_2]\|_2$$

$$= \sum_j \|(\mathbf{W}_1 - \mathbf{W}_2)[\lambda \alpha_{1,j}^T \mathbf{X}_1 - (1-\lambda)\alpha_{2,j}^T \mathbf{X}_2]\|_2 = \text{LHS}$$

Note that

$$LHS = \sum_j \|(\mathbf{W}_1 - \mathbf{W}_2)[\lambda \tilde{\mathbf{x}}_{u,j} - \alpha_{2,j}^T \mathbf{X}_2]\|_2$$

$$= \sum_j \|\mathbf{W}_1 - \mathbf{W}_2[\lambda \tilde{\mathbf{x}}_{u,j} - \lambda \mathbf{x}_{2,\phi_2(j)} + \lambda \mathbf{x}_{2,\phi_2(j)} - \alpha_{2,j}^T \mathbf{X}_2]\|_2$$

$$\leq \sum_j \|\mathbf{W}_1 - \mathbf{W}_2\|_2 (\lambda \|\tilde{\mathbf{x}}_{u,j} - \mathbf{x}_{2,\phi_2(j)}\|_2 + \|\mathbf{X}_2\|_2),$$

where the last inequality follows the triangle inequality and the fact that $0 \leq \lambda \leq 1$. Similarly, we have

$$LHS = \sum_j \|(\mathbf{W}_1 - \mathbf{W}_2)[\alpha_{1,j}^T \mathbf{X}_1 - (1 - \lambda)\tilde{\mathbf{x}}_{u,j}]\|_2$$

$$= \sum_j \|\mathbf{W}_1 - \mathbf{W}_2[\alpha_{1,j}^T \mathbf{X}_1 - (1 - \lambda)\mathbf{x}_{1,\phi_1(j)} + (1 - \lambda)\mathbf{x}_{1,\phi_1(j)} - (1 - \lambda)\tilde{\mathbf{x}}_{u,j}]\|_2$$

$$\leq \sum_j \|\mathbf{W}_1 - \mathbf{W}_2\|_2((1 - \lambda)\|\mathbf{x}_{1,\phi_1(j)} - \tilde{\mathbf{x}}_{u,j}\|_2 + \|\mathbf{X}_1\|_2).$$

In practice, it is easy to normalize all the tasks in the feature space, which leads to $\|\mathbf{X}_1\|_2 = \|\mathbf{X}_2\|_2$ Therefore, by minimizing the EMD loss

$$d_{EMD} = \min\{\min_{\phi_2} \sum_j \|\tilde{\mathbf{x}}_{u,j} - \mathbf{x}_{2,\phi_2(j)}\|_2, \min_{\phi_1} \sum_j \|\tilde{\mathbf{x}}_{u,j} - \mathbf{x}_{1,\phi_1(j)}\|_2\},$$

the proposed task up-sampling network identifies from the candidate set $\{\tilde{T}_u\}$ the task $T_u$ that has the minimal distance between $\mathbf{Y}_u$ and $f_{\theta_u}(\mathbf{X}_u)$; in other words, the task-awareness is maximized.

$\square$

Previous task augmentation approaches directly mix up two tasks without minimizing the EMD loss, i.e., $\mathbf{y}_{u,j} = (1 - \lambda)\mathbf{y}_{1,j} + \lambda\mathbf{y}_{2,j}, \mathbf{x}_{u,j} = (1 - \lambda)\mathbf{x}_{1,j} + \lambda\mathbf{x}_{2,j}$. In this case, the task-awareness is unwarranted as we have illustrated in Section 1, provided that $\|\mathbf{Y}_u - f_{\theta_u}(\mathbf{X}_u)\|_2 = \sum_j \|(1-\lambda)\mathbf{y}_{1,j} + \lambda\mathbf{y}_{2,j} - [(1-\lambda)\mathbf{W}_1 + \lambda\mathbf{W}_2][(1-\lambda)\mathbf{x}_{1,j} + \lambda\mathbf{x}_{2,j}]\|_2 = \sum_j \lambda^2(1-\lambda)^2\|(\mathbf{W}_1 - \mathbf{W}_2)(\mathbf{x}_{1,j} - \mathbf{x}_{2,j})\|_2$.

Table 16: 1-shot meta-training on MiniImangenet-S and meta-testing on various meta-datasets.

| mini-S → | Derm-S | CUB | Aircraft | Fungi | Texture |
|---|---|---|---|---|---|
| MAML | 34.46% | 39.81% | 27.92% | 30.06% | 26.29% |
| MAML+ATU | 36.86%(↑**2.40%**) | 40.67%(↑**0.86%**) | 30.11%(↑**2.19%**) | 32.81%(↑**2.75%**) | 27.28%(↑**2.40%**) |
| MetaSGD | 31.07% | 39.94% | 28.71% | 30.96% | 25.75% |
| MetaSGD+ATU | 37.75%(↑**6.68%**) | 42.52%(↑**2.58%**) | 30.22%(↑**1.51%**) | 32.52%(↑**1.56%**) | 28.61%(↑**2.86%**) |

Table 17: 5-shot meta-training on MiniImangenet-S and meta-testing on various meta-datasets.

| mini-S → | Derm-S | CUB | Aircraft | Fungi | Texture |
|---|---|---|---|---|---|
| MAML | 50.36% | 57.02% | 36.63% | 40.96% | 36.61% |
| MAML+ATU | 51.98%(↑**1.62%**) | 61.04%(↑**4.02%**) | 40.19%(↑**3.56%**) | 43.59%(↑**2.63%**) | 37.60%(↑**0.99%**) |
| MetaSGD | 49.07% | 55.87% | 37.94% | 39.76% | 33.84% |
| MetaSGD+ATU | 54.60%(↑**5.53%**) | 60.37%(↑**4.50%**) | 39.11%(↑**1.17%**) | 42.77%(↑**3.01%**) | 36.59%(↑**2.75%**) |

Table 18: 1-shot meta-training on DermNet-S and meta-testing on various meta-datasets.

| Derm-S → | mini-S | CUB | Aircraft | Fungi | Texture |
|---|---|---|---|---|---|
| MAML | 28.78% | 35.10% | 28.03% | 26.71% | 26.17% |
| MAML+ATU | 30.68%(↑**1.90%**) | 36.37%(↑**1.27%**) | 29.31%(↑**1.28%**) | 27.16%(↑**0.45%**) | 27.11%(↑**0.94%**) |
| MetaSGD | 28.17% | 32.69% | 26.07% | 25.19% | 25.02% |
| MetaSGD+ATU | 30.78%(↑**2.61%**) | 35.86%(↑**3.17%**) | 31.56%(↑**5.49%**) | 28.07%(↑**2.88%**) | 28.04%(↑**3.02%**) |

Table 19: 5-shot meta-training on DermNet-S and meta-testing on various meta-datasets.

| Derm-S → | mini-S | CUB | Aircraft | Fungi | Texture |
|---|---|---|---|---|---|
| MAML | 41.29% | 53.44% | 38.30% | 35.04% | 37.01% |
| MAML+ATU | 46.27%(↑**4.98%**) | 54.99%(↑**1.55%**) | 41.22%(↑**2.92%**) | 35.45%(↑**0.41%**) | 39.04%(↑**2.03%**) |
| MetaSGD | 41.83% | 52.32% | 37.27% | 36.74% | 35.27% |
| MetaSGD+ATU | 44.01%(↑**2.18%**) | 58.52%(↑**6.20%**) | 43.28%(↑**6.01%**) | 38.28%(↑**1.54%**) | 38.28%(↑**3.01%**) |

Table 20: Computational cost analysis.

| | Pre-train | Ordinary training | Total |
|---|---|---|---|
| MAML | – | 35,936 s | 35,936 s |
| Ours | 13,512 s | 45,926 s | 59,438 s |

## F   More experiments under the cross-domain setting.

For a more detailed analysis of how the model behaves in a cross-domain setting, we conduct more experiments meta-tested on meta-datasets, as shown in Table 16, 17, 18, 19.

## G   Computational cost for the method

The tasks are generated on the fly during meta-training. Our method includes two stages: (1) pre-training the augmentation network and (2) meta-training of the meta-learner and the augmentation network jointly. For fair comparison with MAML which trains for 50k iterations, we pre-train the augmentation network for 10k iterations, and meta-train for 40k iterations.

In summary, our method's (TU) computation cost is **1.65x** of the vanilla MAML. The breakdown of the computation cost is listed in Table 20.

## H   Experiments on the limited meta-datasets.

In order to further valid the effectiveness of the proposed method, we have conducted experiments to evaluate the three suggested baselines (including Baseline++ [7], RFS [30], and ProtoNet [27]) on the setting of limited tasks, and show the comparison results in Table 21. We construct the dataset CUB-S, Fungi-S, Aircraft-S, and Texture-S similarly to miniImagenet-S. The details of their construction are listed as follows.

**CUB-S.** CUB [34] is a bird image dataset including 11,788 photos of 200 bird species. In this paper, we randomly select 48 species with 60 images in each species. We devide them into meta-training/meta-validation/meta-testing sets as 12/16/20 species.

- **Meta-training classes:** *Savannah Sparrow, Dark eyed Junco, Black footed Albatross, Henslow Sparrow, Cape Glossy Starling, Black throated Sparrow, Northern Waterthrush, Hooded Warbler, Baltimore Oriole, Scarlet Tanager, Cerulean Warbler, Downy Woodpecker.*

- **Meta-validation classes:** *Mockingbird, Vermilion Flycatcher, Cape May Warbler, Prothonotary Warbler, White crowned Sparrow, Ovenbird, Pomarine Jaeger, Indigo Bunting, Blue winged Warbler, Chipping Sparrow, Horned Grebe, Fox Sparrow, Green Violetear, Nashville Warbler, Least Tern, Marsh Wren.*

- **Meta-testing classes:** *Rose breasted Grosbeak, Nighthawk, Long tailed Jaeger, Bronzed Cowbird, California Gull, Ivory Gull, Northern Fulmar, Brown Pelican, Ring billed Gull, Great Grey Shrike, White breasted Nuthatch, Mourning Warbler, Sage Thrasher, Horned Puffin, Pied Kingfisher, Shiny Cowbird, Scott Oriole, Red eyed Vireo, Song Sparrow, Winter Wren.*

Table 21: Complete classification results of Table 4 with 95% confidence interval.

| Setting | Model | CUB-S | Fungi-S | Aircraft-S | Texture-S |
|---------|-------|-------|---------|-----------|-----------|
| 1-shot | Protonet [27] | $35.35 \pm 0.70\%$ | $26.01 \pm 0.51\%$ | $30.26 \pm 0.56\%$ | $26.52 \pm 0.53\%$ |
| | Protonet+MLTI | $36.17 \pm 0.72\%$ | $28.80 \pm 0.57\%$ | $33.26 \pm 0.68\%$ | $28.28 \pm 0.56\%$ |
| | **Protonet+TU** | $38.35 \pm 0.71\%$ | $30.91 \pm 0.59\%$ | $34.87 \pm 0.70\%$ | $29.02 \pm 0.57\%$ |
| | Baseline++ [7] | $43.98 \pm 0.84\%$ | $32.97 \pm 0.74\%$ | $36.28 \pm 0.79\%$ | $31.36 \pm 0.58\%$ |
| | RFS [30] | $43.96 \pm 0.82\%$ | $33.05 \pm 0.70\%$ | $33.68 \pm 0.80\%$ | $31.47 \pm 0.59\%$ |
| | MAML | $41.58 \pm 0.90\%$ | $29.63 \pm 0.64\%$ | $34.54 \pm 0.72\%$ | $33.79 \pm 0.69\%$ |
| | MAML+MLTI | $44.77 \pm 0.88\%$ | $31.34 \pm 0.65\%$ | $37.76 \pm 0.73\%$ | $34.51 \pm 0.69\%$ |
| | **MAML+TU** | $47.23 \pm 0.96\%$ | $33.21 \pm 0.68\%$ | $39.79 \pm 0.80\%$ | $34.82 \pm 0.67\%$ |
| | **MAML+ATU** | $\mathbf{48.33 \pm 0.96\%}$ | $\mathbf{33.66 \pm 0.70\%}$ | $\mathbf{41.31 \pm 0.82\%}$ | $\mathbf{35.26 \pm 0.73\%}$ |
| 5-shot | Protonet [27] | $55.30 \pm 0.75\%$ | $34.06 \pm 0.63\%$ | $50.49 \pm 0.66\%$ | $33.37 \pm 0.58\%$ |
| | Protonet+MLTI | $56.69 \pm 0.77\%$ | $34.44 \pm 0.56\%$ | $51.77 \pm 0.64\%$ | $35.34 \pm 0.56\%$ |
| | Protonet+TU | $58.32 \pm 0.76\%$ | $35.56 \pm 0.62\%$ | $52.24 \pm 0.66\%$ | $37.20 \pm 0.59\%$ |
| | Baseline++ [7] | $54.41 \pm 0.75\%$ | $44.49 \pm 0.75\%$ | $45.84 \pm 0.77\%$ | $40.31 \pm 0.61\%$ |
| | RFS [30] | $55.40 \pm 0.74\%$ | $46.55 \pm 0.77\%$ | $47.05 \pm 0.78\%$ | $40.91 \pm 0.60\%$ |
| | MAML | $57.97 \pm 0.85\%$ | $37.10 \pm 0.65\%$ | $43.62 \pm 0.69\%$ | $39.47 \pm 0.63\%$ |
| | MAML+MLTI | $63.89 \pm 0.81\%$ | $45.64 \pm 0.74\%$ | $55.05 \pm 0.72\%$ | $40.62 \pm 0.67\%$ |
| | MAML+TU | $64.41 \pm 0.82\%$ | $46.99 \pm 0.83\%$ | $55.85 \pm 0.70\%$ | $41.38 \pm 0.65\%$ |
| | MAML+ATU | $\mathbf{65.56 \pm 0.80\%}$ | $\mathbf{47.91 \pm 0.80\%}$ | $\mathbf{56.90 \pm 0.71\%}$ | $\mathbf{42.52 \pm 0.62\%}$ |

**Fungi-S.** Fungi [2] dataset contains 1,500 wild mushroom species with over 100,000 fungi images. We select the sepcies with greater than 150 images and then randomly choose 100 species, where each species contains 150 images. We split them into meta-training/meta-validation/meta-testing sets with 12/16/20 species.

- **Meta-training classes:** *Suillus granulatus, Phaeolus schweinitzii, Cystoderma amianthinum, Pycnoporellus fulgens, Psathyrella candolleana, Meripilus giganteus, Phellinus pomaceus, Laccaria laccata, Laccaria proxima, Amanita excelsa, Ganoderma pfeifferi, Clitopilus prunulus.*

- **Meta-validation classes:** *Agaricus impudicus, Daedaleopsis confragosa, Fomitopsis pinicola, Cortinarius anserinus, Mucidula mucida, Trametes versicolor, Stropharia cyanea, Ramaria stricta, Radulomyces confluens, Gliophorus psittacinus, Psathyrella spadiceogrisea, Coprinopsis lagopus, Daedalea quercina, Amanita muscaria, Armillaria lutea, Vuilleminia comedens.*

- **Meta-testing classes:** *Hygrocybe ceracea, Trametes hirsuta, Polyporus tuberaster, Lacrymaria lacrymabunda, Fistulina hepatica, Gymnopus dryophilus, Amanita rubescens, Fuscoporia ferrea, Craterellus undulatus, Tricholoma scalpturatum, Mycena pura, Russula depallens, Bjerkandera adusta, Trametes gibbosa, Tremella mesenterica, Cerioporus varius, Amanita fulva, Xylodon paradoxus, Cuphophyllus virgineus, Cortinarius flexipes.*

**Aircraft-S.** Aircraft [17] is a fine-grained image dataset that contains 102 categories of aircraft. We randomly choose 100 variants with 100 images in each variant and split them into meta-training/meta-validation/meta-testing with 12/16/20 categories respectively.

- **Meta-training classes:** *MD-90, 737-600, A310, An-12, DR-400, Falcon-900, DC-3, Challenger-600, Fokker-70, Cessna-172, 747-400, ERJ-145.*

- **Meta-validation classes:** *737-900, A340-600, 737-800, 737-400, L-1011, A330-200, Gulfstream-V, 737-500, A340-200, ATR-72, MD-11, CRJ-700, EMB-120, Fokker-100, DC-6, 737-700.*

- **Meta-testing classes:** *707-320, PA-28, Cessna-208, F-A-18, DHC-8-300, ERJ-135, Tornado, BAE-146-200, A321, ATR-42, Saab-2000, Tu-134, Fokker-50, A380, MD-80, Gulfstream-IV, Yak-42, 747-100, 767-400, Embraer-Legacy-600.*

**Texture-S.** Texture [8] dataset contains 47 classes with 5640 images in total, where each class has 120 images. We randomly split them into meta-training/meta-validation/meta-testing with 12/7/10 classes respectively.

- **Meta-training classes:** *pitted, woven, crosshatched, crystalline, sprinkled, lacelike, bubbly, marbled, dotted, bumpy, striped, zigzagged.*

- **Meta-validation classes:** *wrinkled, grid, perforated, cobwebbed, honeycombed, cracked, blotchy.*

- **Meta-testing classes:** *fibrous, matted, scaly, chequered, flecked, paisley, braided, polka-dotted, interlaced, meshed.*