# OpenReview forum: "Adversarial Task Up-sampling for Meta-learning"
_NeurIPS.cc/2022/Conference — NeurIPS 2022 Accept_

### Official Review · Reviewer_pk34 · 2022-06-20

**Rating:** 5
**Confidence:** 3
**Soundness:** 3 good
**Presentation:** 3 good
**Contribution:** 3 good

**Summary:**

The paper devises a method for task augmentation for meta-learning with three properties: (a) task aware; (b) task imaginary and c) adaptive.  The primary contribution of the paper is a new task generation method which takes into account various properties for the generation process.

**Questions:**

Provided above.

**Limitations:**

Task augmentation is an important way to mitigate meta-overfitting and the paper provides a methodological contribution for it.  While the method is not simple, it's well-motivated with certain properties. I feel that the main limitation of the paper is less analysis on the cross-domain tasks which is more realistic for few-shot learning (rather than miniImagenet -> miniImagenet). I am on the borderline with the paper, but would be happy to revisit my score if two of the limitations could be addressed: (a) major - would task augmentation work for cross-domain tasks too and some insights around it? ; (b) computational cost for the method - are the tasks generated on the fly during meta-training or are they generated and stored before meta-training?

**Strengths And Weaknesses:**

Strengths:

- The problem is realistic as task augmentation is a good way to mitigate meta-overfitting and cover the range of the task distribution

- The method though not very intuitive is well-motivated and presented well.  I have more comments about the method (see below)

- Preliminary results are covering multiple aspects such as classification, regression and cross-domain tasks


Weakness:

- The empirical results are not super strong and the improvements are marginal in most of the cases. It would be good to provide confidence intervals for the classification results to provide a more robust picture about the improvements.  The gap between MLTI and the proposed method is not too much (except for ISIC).

- While task augmentation intuitively makes sense for similar domain datasets (e.g. miniImagenet), I would like to see some discussion around the impact of task augmentation for cross-domain tasks. While the paper shows some preliminary results on cross-domain tasks, I would like to see some more discussion and insights around it as I believe it's a more realistic scenario for few-shot learning

- The paper does not provide information on the computational cost for the method, which I would like to see. Can you please provide some more details on the generation cost and how you incorporate task augmentation during meta-training?

- The theoretical results seem a little forced. I feel the methodological contribution is quite good and more empirical insights to support the results would be better than the theoretical analysis in this regard.

---

> ### Author Response · Authors · 2022-08-02
> **Response to Reviewer pk34 Q2, Q3, and Q4**
>
> ##### Q2: Significance of the improvement
> > - To prove that our improvement is indeed statistically significant, we have provided detailed standard deviations for the performances in Table 4/5/6 in **Table 14/15/13 of Appendix D.4**, respectively.
> >
> > - Besides, we have evaluated the proposed methods (i.e., ATU and TU) on 5 **traditional FSL datasets**, including the **mini-Imagenet** and 4 datasets, i.e., CUB (Birds), Fungi, Aircraft, and Texture, from the **Meta-Dataset**. The 5-shot classification accuracies shown in the following table offers additional evidences that our proposed ATU method **consistently improves by up to 4% over MAML and up to 2% over MLTI**, which is not marginal for few-shot learning.
> >
> >   | **5-shot** | CUB| Aircraft|Texture| Fungi| MiniImagenet|
> >   |-|-|-|-|-|-|
> >   |MAML |71.25%(0.84%)|70.37%(0.69%)|45.63%(0.62%)| 55.61%(0.85%)|63.02%(0.90%)     |
> >   |MAML+MLTI| 72.62%(0.76%)| 72.99%(0.65%)| 45.98%(0.61%)| 56.18%(0.83%)| 63.52%(0.76%)|
> >   |MAML+TU| 73.37%(0.79%)|73.59%(0.67%)|46.41%(0.66%)|56.38%(0.81%)|63.92%(0.76%)     |
> >   |MAML+ATU|**73.65**%(0.80%)|**74.27**%(0.64%)|**47.65**%(0.64%)|**57.20**%(0.84%)| **64.47**%(0.75%)|
> >
> > - More importantly, the proposed ATU and TU have improved over the baselines, being compatible with multiple backbones, **by a large margin on the regression tasks** shown in Table 2 and Table 3.
> >
> > - In terms of the **cross-domain setting** discussed in the response to Q1, the proposed algorithm via augmenting imaginary tasks **significantly boosts** the meta-generalization capability to out-of-distribution tasks.
>
> ##### Q3: Computational cost for the method
>
> > - The tasks are generated on the fly during meta-training. Our method includes two stages: (1) pre-training the augmentation network and (2) meta-training of the meta-learner and the augmentation network jointly. For fair comparison with MAML which trains for 50k iterations, we pre-train the augmentation network for 10k iterations, and meta-train for 40k iterations.
> >
> > - In summary, our method's computation cost is **1.65x** of the vanilla MAML. The breakdown of the computation cost is listed as follows：
> >
> >   |      | Pre-train | ordinary trainig | Total    |
> >   | ---- |:---------:|:----------------:|:--------:|
> >   | MAML | --        | 35,936 s         | 35,936 s |
> >   | Ours | 13,512 s  | 45,926 s         | 59,438 s |
> >
> > - Nevertheless, we would like to emphasize that our focus is to improve the performance and we will continue to investigate how to tackle this efficiency problem in the future.
>
> ##### Q4: Theoretical results seem a little forced.
>
> > - We sincerely thank the reviewer for acknowledging our methodological contribution. The **motivation of our theoretical analysis** is to prove that minimizing the EMD loss between the up-sampled task $\mathcal{T}_u$ and the groundtruth tasks $\mathcal{T}_g$, as the proposed ATU does, maximizes task-awareness, which is evaluated by $\Vert \mathbf{Y}\_u-f\_{\theta_u}(\mathbf{X}_u)\Vert$.
> >
> > - In our updated theoretical analysis in Line 234-261, we have proved that the proposed TU algorithm indeed maximizes task awareness, since the proposed **TU that minimizes the EMD loss** between generated tasks $\mathcal{T}_u$ and groundtruth tasks $\mathcal{T}_g=\\{T_1,T_2\\}$ eventually **minimizes the upper-bound of $\Vert \mathbf{Y}\_u-f\_{\theta_u}(\mathbf{X}\_u)\Vert$**.
> >
> > - The proof has been conducted under two reasonable assumptions:
> >
> >   - In light of the difficulty in mathematical formulation of a possible (or any) up-sampled task $\tilde{T}\_{u}$ by our up-sampling network, we reasonably assume a simplified way of characterizing an up-sampled task that lies in the local manifold of $\mathcal{T}_g=\\{T_1, T_2\\} $ as $\tilde{\mathbf{y}}\_{u,j} = \boldsymbol{\alpha}^T\_{1,j}\mathbf{Y}\_{1} + \boldsymbol{\alpha}^T\_{2,j}\mathbf{Y}\_{2}, \tilde{\mathbf{x}}\_{u,j} = \boldsymbol{\alpha}^T\_{1,j}\mathbf{X}\_{1} + \boldsymbol{\alpha}^T\_{2,j}\mathbf{X}\_{2}, \forall j$, where each sample is a convex combination of samples from both $T_1$ and $T_2$.
> >
> > - We believe that the current theoretical results have shed light on core aspects of our algorithm, while we leave the open and challenging problem of alleviating these two assumptions for our future works.

---

> > ### Comment · Reviewer_pk34 · 2022-08-08
> > **Response to Authors**
> >
> > I thank the authors for your response. I would like to maintain my score, given the authors have appropriately addressed my comments.

---

> ### Author Response · Authors · 2022-08-02
> **Response to Reviewer pk34 Q1**
>
> We sincerely appreciate your constructive comments on this paper. We detail our response below point by point. Please kindly let us know if our response addresses the issues you raised in this paper.
> ##### Q1:  More discussion and insights around the impact of task augmentation for cross-domain tasks.
> > - We have previously conducted the **cross-domain experiments** for **regression tasks in Table 3** and **classification tasks in Table 6**.
> > - We have also followed the reviewer's valuable suggestion by exploring more cross-domain settings, including
> >   - **(1-shot)**  meta-training on **MiniImangenet-S** and meta-testing on DermS/CUB/Aircraft/Fungi/Texture
> >     | Mini-S $\rightarrow$|DermS|CUB|Aircraft|Fungi|Texture|
> >     | -- | -- | -- | -- | -- | -- |
> >     |MAML|34.46%|39.81%|27.92%|30.06%|26.29%|
> >     |MAML+ATU|36.86% ($\uparrow$ 2.40\%) |40.67%($\uparrow$ 0.86\%)| 30.11%($\uparrow$2.19\%)|32.81%($\uparrow$ 2.75\%)| 27.28%($\uparrow$ 2.40\%)|
> >     |MetaSGD|31.07%|39.94%|28.71%|30.96%|25.75%|
> >     |MetaSGD+ATU|37.75%($\uparrow$ 6.68\%)| 42.52%($\uparrow$ 2.58\%)| 30.22%($\uparrow$ 1.51\%)| 32.52%($\uparrow$ 1.56\%)| 28.61%($\uparrow$ 2.86\%)|
> >
> >   - **(5-shot)**  meta-training on **MiniImangenet-S** and meta-testing on DermS/CUB/Aircraft/Fungi/Texture
> >
> >     |Mini-S $\rightarrow$|DermS|CUB| Aircraft|Fungi|Texture|
> >     |-|-|-|-|-|-|
> >     |MAML|50.36%|57.02%|36.63%|40.96%|36.61%|
> >     |MAML+ATU|51.98%($\uparrow$ 1.62\%)|61.04%($\uparrow$ 4.02\%)|40.19%($\uparrow$ 3.56\%)|43.59%($\uparrow$ 2.63\%)|37.6%($\uparrow$ 0.99\%)|
> >     |MetaSGD|49.07%|55.87%|37.94%|39.76%|33.84%|
> >     |MetaSGD+ATU|54.60%($\uparrow$ 5.53\%) |60.37%($\uparrow$ 4.50\%)| 39.11%($\uparrow$ 1.17\%)$|42.77%($\uparrow$ 3.01\%)|36.59%($\uparrow$ 2.75\%)|
> >
> >   - **(1-shot)** meta-training on **Derm-S** and meta-testing on Mini-S/CUB/Aircraft/Fungi/Texture
> >
> >     |DermS $\rightarrow$|Mini-S|CUB|Aircraft|Fungi|Texture|
> >     |-|-|-|-|-|-|
> >     |MAML| 28.78%|35.10%|28.03%|26.71%|26.17%|
> >     |MAML+ATU| 30.68%($\uparrow$1.90\%)|36.37%($\uparrow$ 1.27\%)| 29.31%($\uparrow$ 1.28\%)| 27.16%($\uparrow$ 0.45\%)|27.11%($\uparrow$ 0.94\%)|
> >     |MetaSGD|28.17%| 32.69%|26.07%|25.19%|25.02%|
> >     |MetaSGD+ATU|30.78%($\uparrow$ 2.61\%) | 35.86%($\uparrow$ 3.17\%) | 31.56%($\uparrow$ 5.49\%) | 28.07%($\uparrow $2.88\%) | 28.04%($\uparrow$ 3.02\%)|
> >
> >   - **(5-shot)**  meta-training on **Derm-S** and meta-testing on Mini-S/CUB/Aircraft/Fungi/Texture
> >
> >     |DermS $\rightarrow$ |Mini-S| CUB|Aircraft|Fungi|Texture|
> >     |-|-|-|-|-|-|
> >     |MAML|41.29%|53.44%|38.30%|35.04%|37.01%|
> >     |MAML+ATU|46.27%($\uparrow$ 4.98\%)|54.99%($\uparrow$ 1.55\%)|41.22%($\uparrow$ 2.92\%)|35.45%($\uparrow$ 0.41\%)|39.04%($\uparrow$ 2.03\%)|
> >     |MetaSGD|41.83%|52.32%|37.27%|36.74%|35.27%|
> >     |MetaSGD+ATU|44.01%($\uparrow$ 2.18\%) | 58.52%($\uparrow$ 6.20\%) | 43.28%($\uparrow$ 6.01\%)| 38.28%($\uparrow$ 1.54\%) | 38.28%($\uparrow$ 3.01\%)|
> >
> > - We would like to share our insights and discussions for the above results.
> >
> >   - **Distance of meta-testing domains to the meta-training domain**: a well-recognized fact is that it is more challenging to generalize to a domain that is more distant to the meta-training domain. Surprisingly, the proposed ATU shows strong robustness and even higher performance improvement in face of a large domain gap (e.g., the phase change in Table 3 and DermS -> Mini-S/CUB/Aircraft), which is highly preferred. We attribute this to the "task-imaginary" and "task-aware" property of ATU, with which ATU foresees a large number of unseen but correct tasks that help with meta-generalization.
> >
> >   - **Influence of backbones**: we surprisingly observe that ATU can be more effective combined with a stronger backbone like MetaSGD. This can be explained by the facts that (1) augmented tasks usually pose challenges for the meta-learner and (2) ATU is adversarially trained to generate challenging tasks for the current meta-learner. A stronger backbone like MetaSGD (1) makes better use of the augmented tasks and (2) enforces the generation of "real challenging" tasks instead of noisy ones due to the poor performance of the backbone.
> >
> >   - **Influence of shots**: the proposed ATU tends to improve the backbones more under the 5-shot settings. The reason lies in that ATU under the 5-shot settings has access to more groundtruth tasks for training the up-sampling network, so that the underlying task manifold is better learned with a resulting larger improvement.

---

> ### Author Response · Authors · 2022-08-05
> **We would love to hear back from Reviewer pk34.**
>
> Hi Reviewer pk34,
> > We would like to follow up to see if our response addresses your concerns or if you have any further questions. We would really appreciate the opportunity to discuss this further if our response has not already addressed your concerns. Thank you again!

---

### Official Review · Reviewer_iHfP · 2022-06-25

**Rating:** 5
**Confidence:** 3
**Soundness:** 3 good
**Presentation:** 3 good
**Contribution:** 2 fair

**Summary:**

In this paper, an adversarial task up-sampling method is developed to augment the meta-training tasks. The task up-sampling network is trained by minimizing the EMD loss between the generated tasks and ground truth tasks. An adversarial loss based on the gradient similarity between the support and query sets is incorporated into the proposed framework to generate difficult meta-training tasks. Experiment results indicate that the proposed method leads to improved results in few-shot regression and classification tasks.

**Questions:**

Why the experiments are not conducted on more “traditional” (and larger) FSL datasets? For example, it would be nice to see results in traditional miniImagenet, tieredImagerNet, or metaDataset. Most SOTA few-shot learning methods report results on those datasets. Once I see good performance on those traditional FSL datasets, I am willing to raise my score.

**Limitations:**

The authors discussed the limitations in the paper.

**Strengths And Weaknesses:**

Strengths:

The proposed method makes the distribution of generated tasks close to the ground truth task distribution by minimizing the EMD loss.

Utilize adversarial training to generate more difficult tasks.

The proposed method leads to obvious improvements in sinusoidal regression tasks.

Weaknesses:

In few-shot image classification, the ground truth tasks are constructed by mixup of original images. It is known that mixup itself is a data augmentation method and leads to improved performance in various learning tasks. It is not clear why this mixup step is necessary. This mixup step makes it difficult to analyze the effectiveness of the up-sampling network in few-shot classification. It is not clear whether the improved performance of the proposed method is due to mixup or up-sampling network? An ablation study based on non-mixup samples is necessary to show the effectiveness of the proposed method.


In table 4, the few-shot classification accuracy is better than MLTI, but the gap between them is not large. Since the standard deviation of the classification accuracy is not given, it is not clear whether the improvement is statistically significant.

Based on the results of TU and ATU in table 4, the adversarial loss only leads to marginal improvement on few-shot classification. Note that “adversarial” is a keyword in the title. Current experiment results cannot support that adversarial up-sampling is truly effective in improving few-shot classification performance.

---

> ### Author Response · Authors · 2022-08-02
> **Response to Reviewer iHfP**
>
> We sincerely appreciate your comments on this paper. You may find our response below for your concerns. We would really appreciate it if you could let us know if you have any further concerns.
> ##### Q1: Ablation study regarding mixup.
> > We have indeed conducted the ablation study to investigate whether our performance improvement is attributed to mixup only or the up-sampling network in **Table 12 of Appendix D.2**. The results substantiate the effectiveness of our up-sampling network, rather than the mixup itself.
> > - **Naive Baseline$^3$** which directly trains with the groundtruth tasks that we use underperforms ours. Concretely, each groundtruth task is constucted by choosing the nearest image in the memory bank to mix for all images in a class of the task $T_i$. Under the miniImagenet-S (5-shot) setting, the results are,
> >   |Naive Baseline$^3$|Ours (TU)|
> >   |-| -|
> >   |53.91%(0.78%)| 56.33%(0.79%)|
> > - **Naive Baseline$^1$** where each augmented task is constructed by randomly choosing one image in the memory bank to mix for each image of the task $T_i$ and **Naive Baseline$^2$** where each augmented task is constructed by randomly choosing one image in the memory bank to mix for all images in a class of the task $T_i$ significantly underperform ours, evidenced by
> >   |Naive Baseline$^1$|Naive Baseline$^2$|Ours (TU)|
> >   |-|-|-|
> >   |53.49%(0.74%)| 50.25%(0.71%)| 56.33%(0.79%)|
> ##### Q2: Statistical significance for the performances in Table 4
> > - To prove that our improvement is indeed statistically significant, we have provided detailed standard deviations for the performances in Table 4 in **Table 14 of Appendix D.4**.
> > - We have also shown the standard deviations for the experiments with different backbones and under the cross-domain setting in **Table 15 and Table 13 of Appendix D.4**, respectively.
> ##### Q3: Experiments on traditional FSL datasets.
> > - Our current experiments on the few-shot learning datasets in a limited size follow the setting adopted in the SOTA method MLTI, given that task augmentation is the **most contributory to settings where only a limited size of tasks is accessible**.
> > - We have followed the reviewer's great suggestion and conducted experiments on 5 **traditional FSL datasets**, including the **mini-Imagenet** and 4 datasets, i.e., CUB (Birds), Fungi, Aircraft, and Texture, from the **Meta-Dataset**.
> >   - The results in the following two tables show that our ATU method still improves by up to **4% over MAML and up to 2% over MLTI**, which further validates the effectiveness of our method.
> >   - Generally speaking, the proposed TU/ATU under the 5-shot setting has access to more groundtruth tasks for training the up-sampling network, so that the underlying task manifold is better learned with a resulting larger improvement.
> >
> > - It is worth noting that our method as a task augmentation method, **being orthogonal to recent few-shot learning contributions**, can be combined  with the existing SOTA FSL methods for further improvement.
> >
> > | **5-shot** |CUB| Aircraft|Texture|Fungi| MiniImagenet|
> > |-|:-:|:-:|:-:|:-:|:-:|
> > |MAML|71.25%(0.84%)|70.37%(0.69%)|45.63%(0.62%)|55.61%(0.85%)|63.02%(0.90%)|
> > |MAML+MLTI| 72.62%(0.76%)|72.99%(0.65%)|45.98%(0.61%)|56.18%(0.83%)|63.52%(0.76%)|
> > |MAML+TU|73.37%(0.79%)|73.59%(0.67%)|46.41%(0.66%)|56.38%(0.81%)|63.92%(0.76%)|
> > |MAML+ATU|**73.65**%(0.80%)|**74.27**%(0.64%)|**47.65**%(0.64%)|**57.20**%(0.84%)| **64.47**%(0.75%)|
> >
> > |**1-shot**|CUB|Aircraft|Texture|Fungi|MiniImagenet|
> > |-|:-:|:-:|:-:|:-:|:-:|
> > |MAML|55.96%(1.10%)|52.40%(0.32%)|31.40%(0.27%)|41.27%(0.30%)|47.17%(1.20%)|
> > |MAML+MLTI| 56.68%(0.97%)| 54.73%(0.91%)| 32.62%(0.70%)|42.57%(0.93%)|47.38%(0.70%)|
> > |MAML+TU| 57.66%(0.99%)| 54.97%(0.88%)| 33.00%(0.67%)|42.89%(0.88%)| 47.54%(0.72%)     |
> > |MAML+ATU| **58.76**%(1.10%)|**56.53**%(0.94%)|**33.60**%(0.68%)|**43.82**%(0.89%)| **47.86**%(0.71%)|
> >
> > - The performance gap between the original MAML (48.7% and 63.1%) and our reported results in the table in (47.17% and 63.02%) is caused by the randomness in selecting the meta-testing tasks. We re-run their code in our server and get the reported results. For a fair comparison, though, all baselines and ours (TU/ATU) are evaluated on the same server so that the same set of meta-testing tasks is used.
>
> ##### Q4: Marginal improvement of the adversarial loss on few-shot classification
> > - The results on traditional FSL datasets (i.e., CUB/Fungi/Aircraft/Texture) shown above offer evidence that ATU **consistently improves by 1~2%** over TU, suggesting that the performance gains from the adversarial loss are not marginal on the traditional FSL datasets.
> > - The confidence interval for Table 4 detailed in Table 14 of Appendix D.4 validates the **consistent improvement** by the adversarial loss.
> > - Importantly, the results on the regression tasks shown in Table 2 and Table 3 demonstrate the significant improvement offered by the adversarial loss.

---

> ### Author Response · Authors · 2022-08-05
> **We would love to hear back from Reviewer iHfP.**
>
> Hi Reviewer iHfP,
> > We would like to follow up to see if our response addresses your concerns or if you have any further questions. We would really appreciate the opportunity to discuss this further if our response has not already addressed your concerns. Thank you again!

---

> ### Comment · Reviewer_iHfP · 2022-08-05
> **After rebuttal**
>
> Thank you for the reply from the authors.
>
> I am convinced that the up-sampling network improves performance based on the ablation studies.
>
> However, I am concerned about the performance of the proposed method for few-shot learning in general. Based on the performance of traditional FSL datasets, the proposed method is better than MAML and MLTI but weaker than simple baseline methods [1, 2], like Baseline++. Then the proposed method is only useful in settings where a limited size of tasks are available.
>
> If the proposed method is specifically designed for settings with "a limited size of tasks ", then the authors have to show that the proposed method outperforms those baseline FSL methods [1,2 3] in those settings. In Tables 2, 4, 5, and 6, I just see the comparison to MAML-family methods. Based on my experience, some baseline methods [1, 2, 3] are more robust and powerful than MAML on various datasets, while being much simpler. If the proposed method cannot outperform baseline methods [1, 2, 3], I cannot be convinced by the effectiveness of the proposed methods in settings where a limited size of tasks are available.
>
> I notice that some papers get published without comparing to simple baseline methods. It does not mean that it is the correct way of doing experiments.
>
> [1] Chen, Wei-Yu, et al. "A closer look at few-shot classification." ICLR 2020.
>
> [2] Tian, Yonglong, et al. "Rethinking few-shot image classification: a good embedding is all you need?." European Conference on Computer Vision. Springer, Cham, 2020.
>
> [3] Snell, Jake, Kevin Swersky, and Richard Zemel. "Prototypical networks for few-shot learning." Advances in neural information processing systems 30 (2017).

---

> > ### Author Response · Authors · 2022-08-08
> > **Response to Reviewer iHfP about Comparison with Simple Baseline Methods**
> >
> > Thank you so much for letting us know your remaining concern so quickly. Below please kindly find our response to the remaining concern, and will definitely follow the reviewer's great suggestion to include the results on the suggested baselines. Please kindly let us know if you have any further issues.
> >
> > > We have conducted experiments to evaluate the **three suggested baselines** (including Baseline++ [1], RFS [2], and ProtoNet [3]) on **the setting of limited tasks**, and show the comparison results in the following table. We summarize the major conclusions as below.
> > >
> > > - Performance comparison between **vanilla MAML** and Baseline++ [1], RFS [2], and Protonet [3]
> > >
> > >   - The vanilla MAML is indeed not as competitive as Baseline++ and RFS, as the reviewer expected.
> > >
> > >   - The vanilla MAML outperforms Protonet, which is aligned with Table 3 in the work of MLTI [Yao et al. 2022]. Furthermore, we have investigated the reason by calculating the central kernel alignment (CKA) feature similarity between the model meta-trained on the full dataset and that meta-trained on the limited size dataset. Figure 11 in the revised Appendix F shows that **Protonet is most sensitive to the number of tasks** with the largest drop in feature similarity, especially in the latter layers.
> > > - Performance comparison between **MAML+TU/ATU (ours)** and Baseline++ [1], RFS [2], and Protonet [3]
> > >   - MAML+TU/ATU significantly boosts the performance of vanilla MAML, thereby **outperforming the three suggested baselines by a large margin**. Concretely, MAML+ATU outperforms the best of the three baselines by 9.8% (18.3%) / 1.8% (2.9%) / 12.6% (12.7%) / 12% (3.9%) / 4.3% (3%) on the CUB-S / Fungi-S / Aircraft-S / Texture-S / Mini-S datasets under the 1-shot (5-shot) setting.
> > > - Performance comparison between **Protonet+TU** and Protonet [3]
> > >   - As Protonet is also based on episodic training, the proposed TU/ATU is also applicable to it. The proposed TU/ATU demonstrates great **compatibility with Protonet**, and its **effectiveness** by improving Protonet itself by 8.5% (5.5%)/ 18.8% (4.4%)/ 15.2% (3.5%) / 9.4% (11.5%) / 16.7% (11.3%) on the CUB-S / Fungi-S / Aircraft-S / Texture-S / Mini-S datasets under the 1-shot (5-shot) setting.
> > >
> > > | （1-shot) |CUB-S|Fungi-S| Aircraft-S|Texture-S|Mini-S|
> > > |-|-|-|-|-|-|
> > > |Protonet[3]|35.35%(0.70%)|26.01%(0.51%)|30.26%(0.56%)|26.52%(0.53%)|36.26%(0.70%)|
> > > |Protonet + MLTI|36.17%(0.72%)|28.80%(0.57%)|33.26%(0.68%)|28.28%(0.56%)|41.09%(0.76%)|
> > > |**Protonet + TU**|38.35%(0.71%)|30.91%(0.59%)|34.87%(0.70%)|29.02%(0.57%)|42.33%(0.81%)|
> > > |Baseline++[1]|43.98%(0.84%)|32.97%(0.74%)|36.28%(0.79%)|31.36%(0.58%)|40.86%(0.79%)|
> > > |RFS[2]|43.96%(0.82%)|33.05%(0.70%)|36.68%(0.80%)|31.47%(0.59%)|40.80%(0.80%)|
> > > |MAML|41.58%(0.90%)|29.63%(0.64%)|34.54%(0.72%)|33.79%(0.69%)|38.27%(0.74%)|
> > > |MAML+MLTI|44.77%(0.88%)|31.34%(0.65%)|37.76%(0.73%)|34.51%(0.69%)|41.58%(0.72%)|
> > > |**MAML+TU**|47.23%(0.96%)|33.21%(0.68%)|39.79%(0.80%)|34.82%(0.67%)|42.16%(0.76%)|
> > > |**MAML+ATU**|**48.33**%(0.96%)|**33.66**%(0.70%)|**41.31**%(0.82%)|**35.26**%(0.73%)|**42.60**%(0.77%)|
> > >
> > > |(5-shot)|CUB-S|Fungi-S|Aircraft-S|Texture-S|Mini-S|
> > > |-|-|-|-|-|-|
> > > |Protonet[3]|55.30%(0.75%)|34.06%(0.63%)|50.49%(0.66%)|33.37%(0.58%)|50.12%(0.73%)|
> > > |Protonet+MLTI|56.69%(0.77%)|34.44%(0.56%)|51.77%(0.64%)|35.34%(0.56%)|55.11%(0.74%)|
> > > |**Protonet+TU**|58.32%(0.76%)|35.56%(0.62%)|52.24%(0.66%)|37.20%(0.59%)|55.78%(0.69%)|
> > > |Baseline++[1]|54.41%(0.75%)|44.49%(0.75%)|45.84%(0.77%)|40.31%(0.61%)|53.35%(0.76%)|
> > > |RFS[2]|55.40%(0.74%)|46.55%(0.77%)|47.05%(0.78%)| 40.91%(0.60%)|55.12%(0.78%)|
> > > |MAML|57.97%(0.85%)|37.10%(0.65%)|43.62%(0.69%)|39.47%(0.63%)|52.14%( 0.65%)|
> > > |MAML+MLTI|63.89%(0.81%)|45.64%(0.74%)|55.05%(0.72%)|40.62%(0.67%)|55.22%( 0.76%)|
> > > |**MAML+TU**|64.41%(0.82%)|46.99%(0.83%)|55.85%(0.70%)|41.38%(0.65%)|56.33%( 0.69%)|
> > > |**MAML+ATU**|**65.56**%(0.80%)|**47.91**%(0.80%)|**56.90**%(0.71%)|**42.52**%(0.62%)|**56.78**%(0.73%)|
> > >
> > > We also detail the implementation details for the reviewer's reference.
> > >
> > > - CUB-S/Fungi-S/Aircraft-S/Texture-S/Mini-S:  We construct the dataset CUB-S/Fungi-S/Aircraft-S/Texture-S similar to Mini-S with fixed 12 randomly selected meta-training classes.
> > > - Hyperparameters for Baseline++ and RFS: We follow the best practice in the two papers, and also perform grid search of the hyperparameters under this limited setting. The reported results in the table are with the best hyperparameters.
> > >
> > > [1]  Chen, Wei-Yu, et al. "A closer look at few-shot classification." ICLR 2020.
> > >
> > > [2]  Tian, Yonglong, et al. "Rethinking few-shot image classification: a good embedding is all you need?." ECCV, 2020.
> > >
> > > [3]  Snell, Jake, Kevin Swersky, and Richard Zemel. "Prototypical networks for few-shot learning." NIPS (2017).

---

> ### Comment · Reviewer_iHfP · 2022-08-09
> **Increase my score**
>
> Thank you for the additional experimental results. I am convinced that the proposed method is effective when the number of tasks is limited. I will increase my score.

---

### Official Review · Reviewer_Jet1 · 2022-07-11

**Rating:** 6
**Confidence:** 4
**Soundness:** 2 fair
**Presentation:** 3 good
**Contribution:** 2 fair

**Summary:**

The paper proposes to augment tasks in a meta-learning setup with tasks through an adversarial task up-sampling network. The task-upsampling network first generates coarse level task samples that are refined further to get multiple task samples within the coarse sample’s vicinity. The task up-sampling network is trained by including an adversarial loss to generate tasks that maximally contribute to the meta-learner. The authors validate the resulting adversarial task up-sampling network on a synthetic sine regression task and multiple image classification datasets.


**Questions:**

A classification task typically consists of samples and labels. It is unclear how the labels are generated in the up-sampling process.

What is the role of g_s() in the upsampling network of the classification task? The description in lines 207-217 suggests that the residuals on the images directly are passed through the attention network.

Table 6: why did you choose the cross-domain setting where the target task has more samples to learn from? How did the models behave when the two domains were comparable (1-shot setting)?


**Limitations:**

adequately addressed

**Strengths And Weaknesses:**

Strengths

Task augmentation is a popular approach to prevent over-fitting and memorization in meta-learning. The authors propose a task augmentation technique, where the tasks are generated adaptively. The method is interesting, novel, but quite complicated.

The authors have experimented with many benchmarking datasets and techniques to show the effectiveness of the approach. While there is a marginal improvement over existing methods, the improvement is consistently seen across the datasets. The ablations and experiments to analyze the model’s behavior are satisfactory.

The paper is well written and clear to understand.

Weakness

The problem statement is not novel, and the solution is an incremental contribution to the existing literature. As rightly discussed by the authors, there are many works on task augmentation for meta-learning. The manuscript reads like yet another method on this topic.

I find the theoretical analysis to be a bit disconnected from the algorithm with regards to the two prerequisites. The authors have also acknowledged it in the conclusion section.

---

> ### Author Response · Authors · 2022-08-02
> **Response to Reviewer Jet1**
>
> We thank the reviewer for the valuable feedback. We address your concerns below point by point. Please kindly let us know whether you have any further concerns.
> ##### Q1: Label generation for classification tasks
> > - We would first highlight the difference of **the labels of a classification task in episodic-based meta-training** from those of a classification task in standard supervised training. Concretely,
> >   - the label $y$ in episodic-based meta-training is a random chosen value from $\\{0,1,\cdots, N-1\\}$ for an $N$-way classification task;
> >   - labels of the same semantic class are **mutually-exclusive** across different tasks [1,2].
> > - In view of the fact that the label $y$ is not semantically meaningful, we just (1) generate $N$ distinct classes of images for an $N$-way classification task during the up-sampling process, and (2) randomly assign a unique value from $\\{0,1,\cdots, N-1\\}$ to each class of a generated task.
> > - Concretely, as stated in Line 206-210, the process of generating $N$ distinct classes of images follows by (1) reshaping a groundtruth task into a task pool of $K^s+K^q$ tasks, each of which is $N$-way 1-shot and represented as $[x_1,x_2,...,x_N]\in \mathbb{R}^{Nd}$ where $d$ is the dimension of each image, (2) up-sampling the task pool by our up-sampling network, and (3) re-shaping the generated task pool of size $r(K^s+K^q)\times(Nd)$ back to a task containing $rK^s$ support and $rK^q$ query samples each of which is in $N$ classes, where $r$ is the up-sampling ratio.
> >
> > [1] "Meta-learning without memorization." ICLR, 2020.
> >
> > [2] "Improving generalization in meta-learning via task augmentation." ICML.2021.
> ##### Q2: The role of $g_s()$ in the classification task
> > Please kindly check the detailed up-sampling network architecture for the classification task in **Fig 8 of Appendix B**. We have also updated Line 215-218 by introducing $g_s()$. Considering the extremely complex task distribution in classification tasks, we rely on a set of extra $K_M$ residual images as a source of more informative perturbance. Specifically, for each input image $x_i$, we generate the image by $x^u_i=x_i+x_i^{res}$. Specifically,
> > - The residual image $x_i^{res}$ is an attentive average of the $K_M$ residual images, where the attention weights are learned by the attention network.
> > - The attention network takes the set features of $K_M$ residual images as input.
> > - The set features of $K_M$ residual images are obtained via $g_s()$.
> ##### Q3: The cross-domain setting
> > There might be some misunderstanding about the presentation of Table 6.
> > - 1-shot under mini-S and 5-shot under Derm-S are **not indicative of** that we have meta-trained mini-S under the 1-shot setting and conducted meta-testing of Derm-S under the 5-shot setting, or vice versa.
> > - We have actually adopted the same number of shots during meta-training of one domain and meta-testing of the other domain in Table 6, so that 1-shot and 5-shot indicate that we have evaluated these two settings for each cross-domain generalization, i.e., mini-S (1-shot) $\rightarrow$ Derm-S (1-shot) and mini-S (5-shot) $\rightarrow$ Derm-S (5-shot), or vice versa.
> ##### Q4: The prerequisites in the theoretical analysis are a bit disconnected
> > - The two pre-requisites define **an ideal task-aware up-sampling method** that is expected to meet; mathematically, these two pre-requisites signify that the more task-aware a generated task is, the **smaller the value of $\Vert \mathbf{Y}\_u-f_{\theta_u}(\mathbf{X}_u)\Vert$** is.
> > - In our updated theoretical analysis in Line 234-261, we have proved that the proposed TU algorithm indeed maximizes task awareness, since the proposed **TU that minimizes the EMD loss** between generated tasks $\mathcal{T}_u$ and groundtruth tasks $\mathcal{T}_g=\\{T_1,T_2\\}$ eventually **minimizes the upper-bound of $\Vert \mathbf{Y}_u-f\_{\theta\_u}(\mathbf{X}_u)\Vert$**. The proof has been conducted under two reasonable assumptions:
> >   - In light of the difficulty in mathematical formulation of a possible (or any) up-sampled task $\tilde{T}_{u}$ by our up-sampling network, we reasonably assume a simplified way of characterizing an up-sampled task that lies in the local manifold of $\mathcal{T}\_g=\\{T\_1, T_2\\}$ as $\tilde{\mathbf{y}}\_{u,j} = \boldsymbol{\alpha}^T\_{1,j}\mathbf{Y}\_{1} + \boldsymbol{\alpha}^T\_{2,j}\mathbf{Y}\_{2}, \tilde{\mathbf{x}}\_{u,j} = \boldsymbol{\alpha}^T\_{1,j}\mathbf{X}\_{1} + \boldsymbol{\alpha}^T\_{2,j}\mathbf{X}\_{2}, \forall j$, where each sample is a convex combination of samples from both $T_1$ and $T_2$.
> >   - The function $g$ that relates groundtruth tasks in the first prerequisite is a linear combination, i.e., $\theta_u=\lambda\theta_1 + (1-\lambda)\theta_2$.
> > - We believe that the current theoretical results have shed light on core aspects of our algorithm, while we leave the open and challenging problem of alleviating these two assumptions for our future works.

---

> > ### Comment · Reviewer_Jet1 · 2022-08-09
> > **Thank you for the detailed response**
> >
> > I thank the authors for the detailed response.
> >
> > After having read the other reviews and responses, while there is merit in the proposed method resulting in performance improvements, the method is not very novel. Having said that, the paper will pique the interest of meta-learning few-shot learning research community.

---

> > > ### Author Response · Authors · 2022-08-09
> > > **Thank you Reviewer Jet1**
> > >
> > > We are truly grateful for your support and acknowledgement of our contributions in the meta-learning few-shot learning research community, i.e., being the first generative task augmentation approach that first enjoys task-aware and model-adaptive benefits as summarized in Table 1.

---

### Official Review · Reviewer_GY2p · 2022-07-12

**Rating:** 6
**Confidence:** 4
**Soundness:** 3 good
**Presentation:** 2 fair
**Contribution:** 3 good

**Summary:**

The submission proposes a new task augmentation method.
Proposed approach trains a task proposal network that the output task distribution is matched with true task distribution and challenging for current meta-learner.
Experimental results show improved generalization performance on both regression and classification tasks, compared to SOTA task augmentation methods.


**Questions:**

Clarification of weakness 2 (theoretical analysis) and weakness 3 (EMD details).


**Limitations:**

Author addressed the limitation of strong assumptions on their theoretical analysis.

**Strengths And Weaknesses:**

### Comments
The idea of this paper and the experimental support is convincing.
Currently I vote for rejection as I have few unresolved questions, but I'm willing to raise my rating if rebuttal can resolve my questions.

### Strengths
1) The proposed task augmentation method consistently outperforms existing task augmentation methods in classification and regression domain.
2) By introducing task proposal network and appropriate training objective, output task distribution has nice properties (task-aware, task-imaginary, and model-adaptive) which are intuitively helpful for improving task generalization.

### Weakness
1)  The effectiveness of propose method on regression task is only evaluated on sinusoidal regression task, which is still a challenging
benchmark of meta-regression but low-dimensional and synthetic.
As the baseline methods (MLTI[1], MetaMix[2]) have shown, evaluation on higher-dimensional problem will greatly support the strength of
the proposed method over existing methods on regression domain.

2) Theoretical analysis is not understandable.
In my understanding, authors try to prove that minimizing EMD loss is equivalent to minimizing task awareness
$||  \mathbf{Y}_u -  f\_{\theta_u}(\mathbf{X}_u)  ||_2$ under specific assumptions.
However, the proof picks specific solution (Line 239-241) first then upper bound the solution by EMD loss.
Currently I can't understand how the inequality guarantees minimization of task awareness, where the LHS seems to be a known constant according to the assumption.

3) The detail of how to compute Earth Mover Distance (EMD) between two tasks is not clearly described.
This missing detail makes difficult to fully picture how the task up-sampling network is optimized.
Specifically, the detail of how the labels ($\mathbf{Y}$) are treated was a major blank for me as a reader.

## Reference
[1] Yao et al. "Meta-learning with fewer tasks through task interpolation." ICLR2022.
[2] Yao et al. "Improving generalization in meta-learning via task augmentation." ICML2021.


## Post Rebuttal
Thank the authors for their detailed answers.
I now vote for accepting this paper as:
1) all of my questions are resolved
2) provided results on Pose regression task supports the consistency of performance improvement in regression task stronger.

---

> ### Author Response · Authors · 2022-08-02
> **Response to Reviewer GY2p Q1**
>
> We sincerely appreciate your constructive comments on this paper. We detail our response below point by point. Please kindly let us know if our response addresses the issues you raised in this paper.
> ##### Q1: Clarification of our theoretical analysis
> > -  The **motivation of our theoretical analysis** is to prove that minimizing the EMD loss between the up-sampled task $\mathcal{T}_u$ and the groundtruth tasks $\mathcal{T}_g$ , as the proposed ATU does, maximizes task-awareness, which is evaluated by
> $\Vert \mathbf{Y}_u -  f\_{\theta_u}(x_u) \Vert$.
>
> > - Such a theoretical analysis requires the **mathematical formulation of a possible (or any) up-sampled task $\tilde{T}_{u}$** by our up-sampling network, which is not necessarily the optimal one that has the minimal EMD loss. However, this is very challenging, so that we reasonably assume a simplified way of characterizing an up-sampled task that lies in the local manifold of $\mathcal{T}_g=\\{ T_1, T_2\\}$ as $\tilde{\mathbf{y}}\_{u,j} = \boldsymbol{\alpha}^T\_{1,j}\mathbf{Y}\_{1} + \boldsymbol{\alpha}^T\_{2,j}\mathbf{Y}\_{2}$, $\tilde{\mathbf{x}}\_{u,j} = \boldsymbol{\alpha}^T\_{1,j}\mathbf{X}\_{1} + \boldsymbol{\alpha}^T\_{2,j}\mathbf{X}\_{2}$, $\forall j,$  where each sample is a convex combination of samples from both $T_1$ and $T_2$. Note that we have **followed the reviewer's constructive comment and updated the theoretical analysis section (Line 234 - Line 261) and Appendix E**. Please kindly check the full proof in Appendix E.
> >
> >     - The combination coefficients $\boldsymbol{\alpha}\_{1,j},\boldsymbol{\alpha}\_{2,j}\in\mathbb{R}^{(K^s+K^q)\times 1},$ $\sum\_k^{K^s+K^q} {\alpha}\_{1,jk}=1$, $\sum\_k^{K^s+K^q} {\alpha}\_{2,jk}=1$, $\alpha_{1,jk},\alpha_{2,jk}\geq 0$ $\forall k$. Different combination coefficients lead to a set of up-sampled task candidates $\{\tilde{T}_u\}$.
> >  -   Based on the formulation, we evaluate the task-awareness property of each candidate $\tilde{T}\_u$ i.e., the distance between $\tilde{\mathbf{Y}}\_u$ and $f_{\\!\theta\_u}(\tilde{\mathbf{X}}\_u)$. We have proved that by minimizing the EMD loss $d\_{EMD} =\min\\{\min\_{\phi\_2}\sum\_j \Vert\tilde{\mathbf{x}}\_{u,j} - \mathbf{x}\_{2,\phi\_2(j)}\Vert\_2,\min\_{\phi\_1}\sum\_j \Vert\tilde{\mathbf{x}}\_{u,j} - \mathbf{x}\_{1,\phi\_1(j)}\Vert\_2\\}$,the proposed task up-sampling network identifies from the candidate set $\\{\tilde{T}\_u\\}$ the task that has the minimal distance between $\mathbf{Y}\_u$ and $f\_{\theta\_u}(\mathbf{X}\_u)$; in other words, the task-awareness is maximized.

---

> ### Author Response · Authors · 2022-08-02
> **Response to Reviewer GY2p Q2 and Q3**
>
>
> ##### Q2: EMD details
> > - We would like to apologize for the abused notation $x$ in both Eq. (2) and Section 4.1, which causes confusion. We have already **corrected the notation from $x$ to $s$ in the EMD loss in Eq. (2)**, which denotes a point in the set $S_1$. In our problem,
> >   - the set $S_1$ is the set of up-sampled tasks $\mathcal{T}_{up}$.
> >   - the set $S_2$ is the set of ground-truth tasks $\mathcal{T}_{g}$.
> >   - each point $s$ in either set represents the embedding of a task, which we detail below for regression and classification tasks, respectively.
> > - **Embedding of a sine regression task**: each task includes a support set of size $K^s$ and a query set of size $K^q$, and each sample in the support and query sets is characterized both by $(x,y)$ values.
> >    - We combine all samples of the support set and query set as the embedding of a sine regression task, i.e., $s=[x_1^s,y_1^s,...,x_{K^s}^s,y_{K^s}^s,x_1^q,y_1^q,...,x_{K^q}^q,y_{K^q}^q] \in \mathbb{R}^{2(K^s+K^q)}$ where $x_1^s\le x_2^s \le ... \le x_{K^s}^s$ and $x_1^q\le x_2^q \le ... \le x_{K^q}^q$ .
> >   - Consequently, the dimension of generated tasks $\mathcal{T}_{up}$ is $(rN_p,2(K^s+K^q))$ , and that of ground truth tasks $\mathcal{T_g}$ is $(rN_p,2(K^s+K^q))$, where $r$ is the up-sampling ratio and $N_p$ is the set size of a task patch (see Line 143).
> >   - $d\_{EMD}(\mathcal{T}\_{up},\mathcal{T\_g})$ in Eq. (6), therefore, follows the definition of Eq. (2) by substituting the $\mathcal{T}\_{up}$ defined above for $S_1$ and substituting the $\mathcal{T}_{g}$ defined above for $S_2$.
> > - **Embedding of an $N$-way classification task**: each task consists of a support set of size $K^s$ and a query set of size $K^q$. Different from sine regression tasks, the label $y$ in episodic-based meta-training is a randomly chosen value from $\\{0,1,\cdots, N-1\\}$ for each task and labels of the same class are mutually-exclusive across different tasks [1].
> >   - In view of the fact that the label $y$ is not semantically meaningful, we represent the embedding of an $N$-way classification task with images $x$ only. Concretely, we reshape each task into a task pool of $K^s+K^q$ tasks, each of which is $N$-way 1-shot and represented as $s=[x_1,x_2,...,x_N]\in \mathbb{R}^{Nd}$ where $d$ is the dimension of each image.
> >   -  Consequently, the dimension of generated tasks $\mathcal{T}_{up}$ is $(r(K^s+K^q),Nd)$ , and that of ground truth tasks $\mathcal{T_g} $ is $(r(K^s+K^q),Nd)$, where r is the up-sampling ratio, $\mathcal{T}_g$ is obtained by mixing up with images in the memory bank.
> >   -  The computation of $d_{EMD}(\mathcal{T}_{up},\mathcal{T_g})$ in Eq. (5) follows Eq. (2). Note that a generated task can be reshaped back to obtain the support and query set.
> > - We have provided more details on the EMD loss in the revised version. \
> [1] "Meta-learning without memorization." ICLR. 2020.
> ##### Q3: Evaluation on higher-dimensional regression problems
> > - The capability of our method in tackling high-dimensional features has been proved in the image classification problems.
> > - For further proving the effectiveness of the proposed method on higher-dimensional regression problems, we have followed the reviewer's suggestion to implement and evaluate our method for the Pose regression benchmark dataset that has been adopted in MLTI [1] and MetaMix [2]
> >   - The results in the following table show that TU still outweighs the previous baselines MLTI [1] by 4% and achieves the new state-of-the-art.
> >     |      |  MetaMix   |    MLTI    |    Ours     |
> >     | :--: | :--------: | :--------: | :---------: |
> >     | MSE  | 2.09(0.23) | 1.97(0.21) | 1.89 (0.17) |
> >
> > [1] Yao et al. "Meta-learning with fewer tasks through task interpolation." ICLR, 2022.
> >
> > [2] Yao et al. "Improving generalization in meta-learning via task augmentation." ICML, 2021.

---

> ### Author Response · Authors · 2022-08-07
> **We would love to hear back from Reviewer GY2p**
>
> Hi Reviewer GY2p,
>
> > We would like to follow up to see if our response addresses your concerns or if you have any further questions. We would really appreciate the opportunity to discuss this further if our response has not already addressed your concerns. Thank you again!

---

### Comment · Area_Chair_hHhr · 2022-08-10
**Please participate in the discussion**

Dear reviewers,

The interactive author-reviewer discussion period is now over, and we need to have an internal discussion to decide whether to accept or reject the paper. It seems that all of you are leaning toward acceptance. However, some of you seem to have not yet read authors' responses, so please go over them, leave feedback, and read the reviews from the others, and share your final opinion on the paper with the rest of us.

Thanks,
AC

---

### Meta-Review · Area_Chair_hHhr · 2022-08-27

**Recommendation:** Accept
**Confidence:** Certain

**Metareview:**

This paper proposes a task augmentation method for meta-learning that generates new tasks which match the true task distribution and are also challenging for the current meta-learner. This is done by training a task upsampling network with an adversarial loss as well as an EMD loss between the adversarially generated and ground-truth tasks. The authors provide a theoretical analysis that the tasks generated by their adaptive task upsampling framework are indeed task-aware, or comply with the true task distribution, and validate the method on both regression and classification tasks. The results show that the proposed task upsampling method outperforms existing regularization methods or task augmentation methods.

The paper initially received split reviews. Reviewers were generally positive about the introduction of the desirable properties for augmented tasks in meta-learning as a meaningful contribution. They also found the proposed method with adversarial task-aware upsampling as novel and interesting, and the experimental validation as adequately showing the effectiveness of the proposed method as well as each of its components. Another advantage that is not mentioned by the reviewers, is that it is applicable to both regression and classification tasks.

However, reviewers were also concerned with unclear, and somehow disconnected theoretical analysis from the actual framework, marginal improvements over state-of-the-art baselines such as MLTI, unclear effectiveness of the adversarial loss, and missing results on larger few-shot classification datasets. Most of these points were addressed in the author response, which resulted in some of the reviewers raising their scores, and all reviewers leaned toward acceptance after the interactive discussion period.

In sum, this is a well-written paper that introduces meaningful insights about task augmentation in meta-learning, as well as a novel method, and may be of interest to researchers working on the topic. However, method-wise, its relatively weak improvement over existing, simpler task augmentation methods may diminish the potential practical impact of the work. Yet, the advantages outweigh its drawbacks and the work is worth sharing at NeurIPS 2022.

**Award:**

No

---

### Decision · Program_Chairs · 2022-09-14

Accept